# Remote automated delivery of mechanical stimuli coupled to brain recordings in behaving mice

Justin Burdge[1,2,3,4†], Anissa Jhumka[1,2,3†], Ashar Khan[1,2,3], Simon Ogundare[1,2,3], Nicholas Baer[1,2,3], Sasha Fulton[1,2,3], Alexander Kaplan[1,2,3], Brittany Bistis[1,2,3], William Foster[1,2,3], Joshua Thackray[4], Andre Toussaint[1,2,3], Miao Li[5], Yosuke M Morizawa[5], Jake Nazarian[6], Leah Yadessa[1,2,3], Arlene J George[4], Abednego Delinois[1,2,3], Wadzanayi Mayiseni[1,2,3], Noah Loran[1,2,3], Guang Yang[5], David J Margolis[4], Victoria E Abraira[4], Ishmail Abdus-Saboor[1,2,3]*

[1]Zuckerman Mind Brain Behavior Institute, Columbia University, New York, United States; [2]Department of Biological Sciences, Columbia University, New York, United States; [3]Howard Hughes Medical Institute, Chevy Chase, United States; [4]Cell Biology and Neuroscience Department, W.M. Keck Center of Collaborative Neuroscience, Rutgers, The State University in New Jersey, New Brunswick, United States; [5]Department of Anesthesiology, Columbia University, New York, United States; [6]University of Illinois, Champaign, United States

*For correspondence:
ia2458@columbia.edu

†These authors contributed equally to this work

## eLife Assessment

This **important** study describes the development and validation of an Automated Reproducible Mechano-stimulator (ARM), a tool for standardizing and automating tactile behavior experiments. The data supporting the use of the ARM system are **compelling**, and demonstrate that by removing experimenter effects on animals, it reduces variability in various parameters of stimulus application. Moreover, the authors demonstrate that any noise emitted from the ARM does not induce an increased stress state. Once commercially available, the ARM system has the potential to increase experimental reproducibility between laboratories in the somatosentation and pain fields.

**Abstract** The canonical framework for testing pain and mechanical sensitivity in rodents is manual delivery of stimuli to the paw. However, this approach is time-consuming, produces variability in results, requires significant training, and is ergonomically unfavorable to the experimenter. To circumvent limitations in manual delivery of stimuli, we have created a device called the automated reproducible mechanostimulator (ARM). Built using a series of linear stages, cameras, and stimulus holders, the ARM is more accurate at hitting the desired target, delivers stimuli faster, and decreases variability in delivery of von Frey hair filaments. We demonstrate that the ARM can be combined with traditional measurements of pain behavior and automated machine-learning-based pipelines. Importantly, the ARM enables remote testing of mice with experimenters outside the testing room. Using remote testing, we found that mice habituated more quickly when an experimenter was not present, and experimenter presence led to significant sex-dependent differences in paw withdrawal and pain-associated behaviors. Lastly, to demonstrate the utility of the ARM for neural circuit dissection of pain mechanisms, we combined the ARM with cellular-resolved microendoscopy in the amygdala, linking stimulus, behavior, and brain activity of amygdala neurons that encode negative pain states. Taken together, the ARM improves speed, accuracy, and robustness

of mechanical pain assays and can be combined with automated pain detection systems and brain recordings to map central control of pain.

## Introduction

The discovery of new mechanisms for nervous system encoding of pain and for identifying new classes of safe analgesics relies heavily on animal models. Since the 1980s, rodents have surpassed dogs and cats as the predominant model organisms to study pain because of increased genetic access to their nervous system, ease of activating the pain system, and ability to work in a high-throughput manner testing many animals at once (*Mogil, 2009*). Indeed, our basic understanding of the neurobiology of pain, from ion channels and genes in peripheral sensory neurons to synaptic transmission into the spinal cord and integrated networks across the brain, has benefitted tremendously from fundamental research in rodent models. While translational impact has lagged behind basic science discoveries, there are still triumphs to note, such as CGRP monoclonal antibodies that work to block pain in rodents and effectively treat migraine pain in the clinic (*Israel et al., 2018*; *Iannone et al., 2022*). Thus, with continued improvements and innovations in how we deliver and measure the complex state of pain in rodents, additional breakthroughs in basic science and translation should abound.

One acknowledged limitation in delivering both innocuous and noxious mechanical stimuli to rodents is the manual experimenter-driven delivery of stimuli to the rodent paw. For more than 50 years, these stimuli have primarily been the von Frey hair (vFH) filaments that are delivered to the mouse paw from an experimenter below the rodent aiming, poking, and subsequently recording a paw lift (*Frey, 1896*; *Dixon, 1980*; *Chaplan et al., 1994*). This technique requires extensive training before researchers feel confident in producing consistent results. However, this consistency does not hold between researchers, with results among experienced researchers from the same lab varying considerably. A meta-analysis of thermal and mechanical sensitivity testing *Chesler et al., 2002*; *Zumbusch et al., 2024* found that the experimenter has a greater effect on results than the mouse genotype, making data from different individual experimenters difficult to merge. Recent studies utilizing the manual high-speed analysis of withdrawal behavior analysis developed by *Abdus-Saboor et al., 2019*, has reproduced this sizable experimenter effect using the new technique (*Rodríguez García et al., 2024*). Additional work has found that experimenter sex has a significant effect on mechanical sensitivity that appears to be mediated via stress (*Sorge et al., 2014*). Previous attempts to decrease this variability in mechanosensory testing have focused on eliminating variability in stimulus delivery with devices such as electronic von Frey and the dynamic plantar asthesiometer, but both still require an experimenter to be present and are focused on measuring a mechanical force threshold, limiting their ability to measure hyperalgesia (*Möller et al., 1998*; *Raposo et al., 2015*; *Urru et al., 2020*; *Jokinen et al., 2018*). If we could standardize the delivery of mechanical stimuli, this could increase throughput and reduce potential variation in performing mechanical sensory testing in rodents.

Historically, the pain field has focused most of its attention on the sensory site of transduction – the nociceptive peripheral sensory neurons. This focus has uncovered many critical genes and ion channels important for pain signaling that are now targets of therapeutic development (*Chen et al., 2020*; *Cummins et al., 2004*; *Alsaloum et al., 2021*). With this said, the brain is also important for pain – from sensory perception to encoding affective components of pain that negatively alter mood and motivation (*Cai et al., 2018*; *Corder et al., 2019*; *Chiang et al., 2020*; *Starr et al., 2009*; *Zhu et al., 2024*). Combining approaches to deliver painful stimuli with techniques mapping behavior and brain activity could provide important insights into brain-body connectivity that drives the sensory encoding of pain.

Here, we have created a robot arm that delivers mechanical stimuli to the paw of freely behaving mice. We demonstrated that this device, which is controlled by an experimenter using a standard video game controller, delivers stimuli more accurately, quickly, and consistently than well-trained experts. Moreover, the device can be controlled remotely, removing potential experimenter disturbances of animal behavior. Lastly, the robot arm can be used with traditional readouts or machine-learning-based measurements of pain and combines seamlessly with brain recording technologies.

## Results

### Automating mechanical stimulus delivery

Our goal for designing the automated reproducible mechanostimulator (ARM) was to eliminate stimulus variability and allow remote delivery. This required that the device be able to stimulate five freely moving mice with multiple stimuli within a session. To this end, three linear stages were mounted and wired together to allow for controlled and customizable movement of the stimulus along the x-, y-, and z-axis. A final rotational axis was attached to the z-axis to allow for both the controlled application of a brush stimulus and the quick switching of stimuli. 3D-printed mounts were then attached to the z-axis to hold a camera for aiming the stimulus at the mouse paw and to the rotational axis to hold stimuli. A high-speed camera was then mounted on a linear stage along with an infrared light to allow for the tracking of the mouse's withdrawal response. This device was then redesigned so that it could fit on 75 cm×50 cm table, using three 500 frames per second (fps) cameras and a force sensor to aid in stimulus delivery (*Figure 1A and B*).

We controlled the ARM with a custom-built Python code that paired the standard Xbox One controller to the bottom-up camera, with calibrated crosshairs superimposed on the camera's video feed. For the first iteration of the device, three types of stimulus delivery were programmed: a simple sine wave motion function along the z-axis for a cotton swab and pinprick stimuli, a combination of concurrent sine waves along the z- and radial axis for brush stimuli, and a slow increase in stimulus height over 3 s followed by quick retraction for von Frey. Each of these delivery types was designed to replicate the manual delivery of those stimuli. Remote desktop software was then used to allow for control of the device from either across the lab or even across the city (*Video 1*). A habituation program was crafted to be used during habituation sessions and before normal testing to get mice used to the sound and movement of the ARM without stimulus making contact with the mouse (*Video 2*). Noise generated by the ARM was observed to be minimal compared to the background HVAC noise. The ARM was assessed for accuracy in both targeting and the force of stimulus delivery. Five different researchers recruited from the lab delivered 10 pinprick stimuli to stationary targets manually and via the ARM. It was found that the ARM decreased the off-target distance of stimuli by 93.3% (*Figure 1C*), while delivering more consistent stimulus based on analysis of stimulus height with high-speed videography (*Figure 1—figure supplement 1*, *Videos 3–6*). This is an important result, as we find that sometimes inexperienced researchers erroneously miss the mouse paw and unknowingly target another part of the animal, like the belly. Regardless of the level of experience, it is extremely difficult to specifically target the same region of the paw within and across stimulus delivery sessions. The ARM provides the ability to precisely identify and stimulate the desired region in a reproducible manner. This is a major strength when investigating biological phenomena at the level of somatosensory receptive fields.

We performed the next experiment to determine if the ARM produced noise that may stress mice. To test this, the open field test was performed with naïve mice (n=10) 2 feet from the ARM while the ARM either sat silent or ran through its habituation program, producing noise. The mouse's center point movement was then tracked in relation to the chamber, its edges, and center. No significant differences were found in distance traveled, center entrances, center, time in center, and latency to center entrance based on a Student's two-tailed t-test (*Figure 1—figure supplement 1D–G*). Based on this, neither stress nor locomotion differences were detected by this test, indicating the ARM does not induce an increased stress state due to its noise, even in non-habituated mice. Mouse waste getting on mechanical parts was found to be a major concern for the initial version of the device. As part of the redesign, the linear stages were moved out from under the mice to avoid this problem. Despite this problem, the original version of the device has not had any of its stages break down yet. A common problem, though, was that stimulus tips would blunt or break if they hit the mesh of the mesh table, requiring replacement. This has been solved in the latest version through a new feature where the mesh is detected via the force sensor, prompting immediate stimulus withdrawal, avoiding damage.

To test the ARM's performance in delivering traditional von Frey filament stimuli, two external researchers with experience in delivering von Frey stimuli were brought in from the Yang lab to assist with testing. The researchers performed canonical von Frey experiments (*Dixon, 1980*; *Zhou et al., 2018*) and were uninformed as to the goals of the study. Both researchers and the ARM applied stimulus to a force sensor in a manner that mimicked their application to the mouse's paw 10 times with

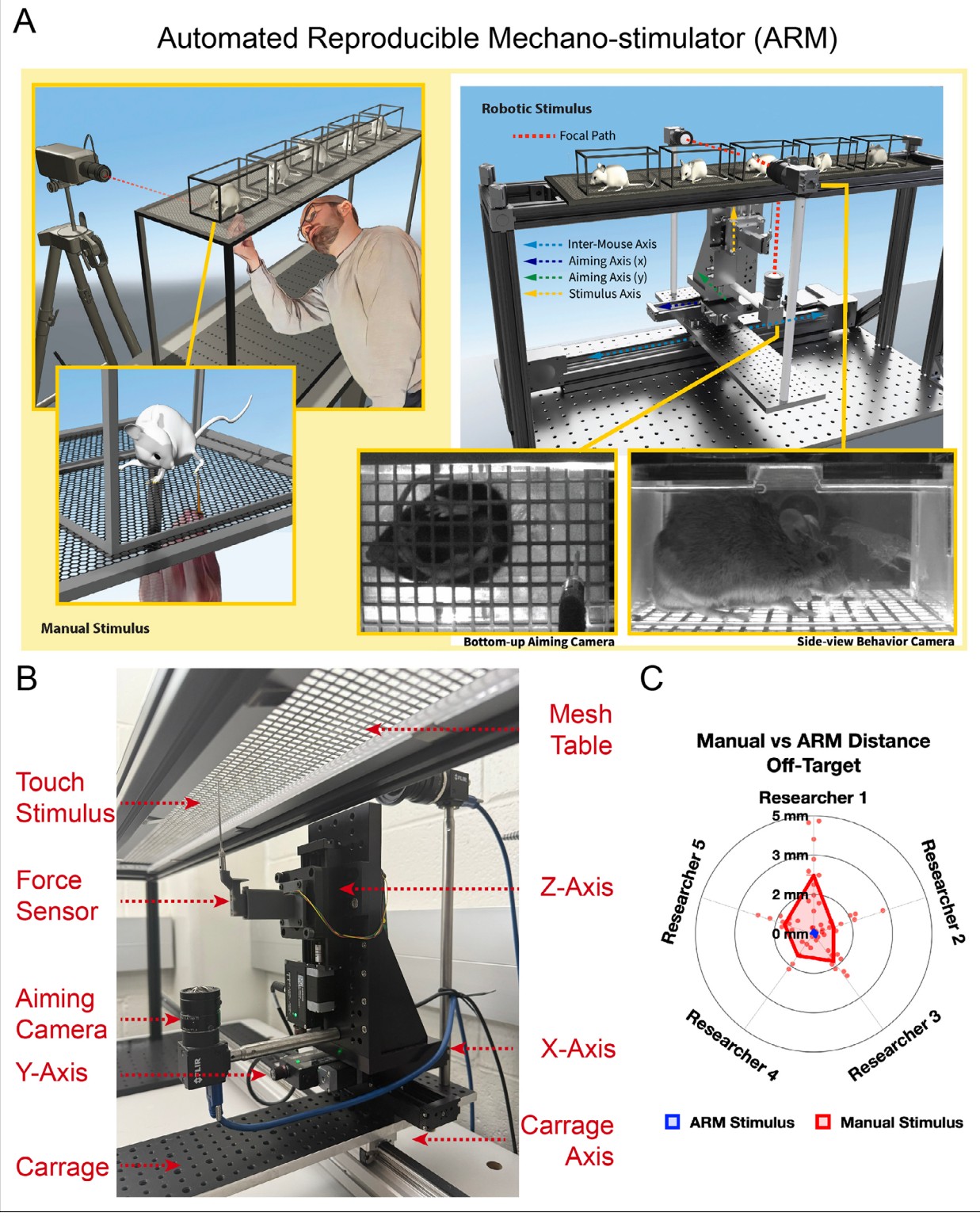

**Figure 1.** Mechanical stimulus delivery with the automated reproducible mechanostimulator (ARM). (**A**) Comparison between manual stimulus delivery that requires a researcher to aim and deliver stimulus by hand in close proximity to mice vs robotic stimulus delivery via the ARM using motorized linear stages to maneuver and deliver stimulus and a bottom camera to aim. (**B**) Zoomed-in schematic showing components of the ARM, including the configuration of the linear axis, the aiming camera to the ARM and the stimulus holder. (**C**) An ARM vs manual stimulus aim comparison was conducted by 5 researchers who delivered 10 instances each of manual and ARM pinprick stimulus to a stationary target. A significant (p<0.0001) 93.3% decrease in distance off-target was observed in ARM stimuli delivery compared to manual delivery.

*Figure 1 continued on next page*

*Figure 1 continued*

The online version of this article includes the following figure supplement(s) for figure 1:

**Figure supplement 1.** Automated reproducible mechanostimulator (ARM) stimulus delivery and comparison with manual delivery.

four filaments before testing a cohort of wild-type mice (n=10) with a range of six von Frey filaments (*Figure 2A*). Both researchers tested the same cohort of mice, and the ARM was used by the third and fourth researchers to test their cohort twice to mimic this. Researchers were instructed to apply vFH stimulus for 2 s, and the ARM was programmed to do the same. Each trial with the force gauge was normalized based on the start time, and the mean and standard deviation of the trial were plotted for each researcher and the ARM. With the same 1.4 g and 2 g von Frey filaments, Researcher 1 delivered max average forces of 1.5 g and 2.7 g, and Researcher 2, 1.35 g and 2.4 g. The ARM delivered average max forces closest to the targeted forces, with 1.36 g and 1.9 g (*Figure 2B and C*). Some of the errors observed could be due to the error rate (±0.05 g) in the force gauge and the von Frey set used. Higher mean standard deviations were observed in the data for the researchers vs the ARM, driven primarily by variation in vFH application time (*Figure 2D*).

To quantify this, the coefficient of variation was calculated for the application time of each vFH (0.6 g, 1 g, 1.4 g, 2 g) for each researcher and the ARM. The coefficient of variation for the researcher's data combined was also calculated to model variation between researchers. The ARM had no variation in stimulus delivery time, whereas manual stimulus delivery had an average 12.4% variation in the stimulus length of individual researchers and 61.17% variation in that of the combined researchers (*Figure 2E*).

Further, we found that the ARM decreased the time needed to apply a stimulus 10 times to a mouse paw by 50.9% compared to manual delivery (*Figure 2—figure supplement 1A*). This effect size may decrease for researchers who leave longer delays between stimulus delivery, but the device should still speed up experiments by reducing aiming time and allowing researchers to quickly switch to a new mouse while waiting for the first. Both manual delivery and the ARM produced significant paw withdrawal percentage curves, a standard traditional measurement of mechanical sensitivity in the field (*Frey, 1896*; *Dixon, 1980*; *Chaplan et al., 1994*; *Figure 2E*), with a two-way ANOVA and a post hoc Tukey test detecting significant increases in comparing the three lower force VFHs (0.02 g, 0.07 g, 0.16 g) to the two highest force VFHs (1 g, 1.4 g). This demonstrates that the ARM delivers results comparable to highly experienced researchers. However, a two-way ANOVA and a post hoc Tukey test found that Researcher 2 elicited a significantly higher ($p=0.0008$) paw withdrawal frequency than Researcher 1 (*Figure 2G*), which corresponded with Researcher 2's higher VFH application time as measured by the force sensor (*Figure 2D*). Comparisons between the third and fourth researchers applying ARM stimulus remotely found no significant differences in response to either individual vFH or the full dataset (*Figure 2H*). Thus, these

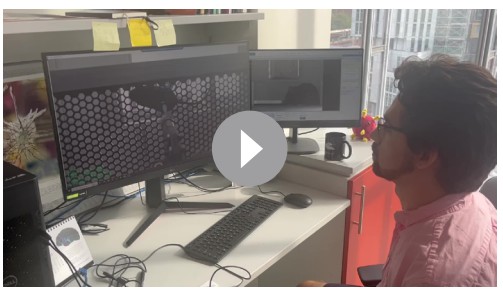

**Video 1.** Remote operation of the automated reproducible mechanostimulator (ARM). Researcher aims the ARM using video feed and crosshairs, switches between stimuli, and delivers stimuli to a fake mouse in the behavior room from their lab bench.

https://elifesciences.org/articles/99614/figures#video1

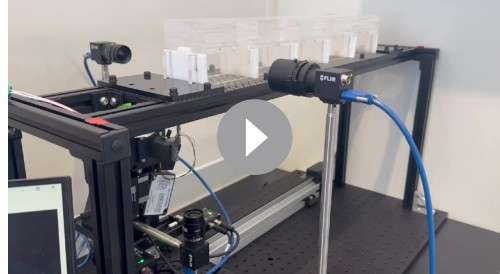

**Video 2.** Automated reproducible mechanostimulator (ARM) with automated habituation program activated, with empty chambers or chamber with a fake mouse for reference. The program lowers the stimulus holder so that stimulus delivery will not cross the mesh. The ARM then moves to the first chamber and randomly moves, pauses, or delivers a stimulus to empty air for 1 min before moving on to the next chamber. This program habituates mice to the presence and noise of the ARM.

https://elifesciences.org/articles/99614/figures#video2

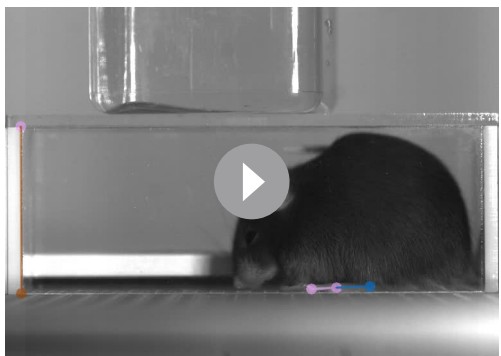

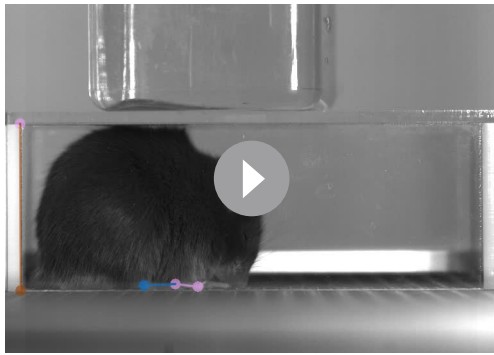

**Video 3.** Representative example of Social LEAP Estimates Animal Poses (SLEAP) tracked 2000 frames per second (fps) cotton swab trial, with manual stimulus delivery. Stimulus is delivered at an angle and lingers at apex. Mouse slowly removes paw from the stimulus and places it back on the mesh.

https://elifesciences.org/articles/99614/figures#video3

**Video 4.** Representative example of Social LEAP Estimates Animal Poses (SLEAP) tracked 2000 frames per second (fps) cotton swab trial, with automated reproducible mechanostimulator (ARM) stimulus delivery. Stimulus is delivered straight up and down, withdrawing quickly after reaching its apex. The mouse slowly removes its paw from the stimulus and places it back on the mesh.

https://elifesciences.org/articles/99614/figures#video4

findings indicate that the ARM decreases variation in VFH stimulus used to measure mechanical sensitivity while decreasing the time needed to perform these assays effectively.

## Expanded ARM system analysis options

An updated version of our lab's pain assessment at withdrawal speeds (PAWS) analysis strategy, along with the ARM's built-in sensors, was used to measure the ARM's effect on evoked pain behavior in mice. Together, this created a system where precise stimulus could be delivered to the mouse paw (*Figure 3A*), automated measures of reflexive features could be taken (*Figure 3B*), and high-speed videos were recorded for analysis of more complex features via PAWS (*Figure 3C*). The ARM's high-speed camera was used to record either 500 fps or 2000 fps videos of the mouse's withdrawal response, and the movement of the mouse's paw in these videos was then tracked using Social LEAP Estimates Animal Poses (SLEAP) (*Pereira et al., 2022*). This tracking data was then fed into the PAWS software to compute measures of reflexive and affective pain behavior (*Figure 3—figure supplement 1A*). More details on the PAWS analysis can be found in *Jones et al., 2020*. To facilitate high-throughput analysis, the PAWS software was updated to allow for simple installation on new devices and a full graphical interface. In addition, features were added to allow for the analysis of SLEAP tracking data, greater control over feature scoring, and support for ARM-assisted measurement of withdrawal latency.

The first version of this withdrawal latency feature facilitated measurement of withdrawal latency to mechanical stimuli using an Arduino to trigger the high-speed camera in response to ARM stimulus delivery. The Arduino was set based on stimuli to trigger the camera 25 ms before the stimulus crossed the mesh and made contact with the paw. Adjustments were then made to the PAWS software to automate the measurement of withdrawal latency based on pose tracking data of the withdrawal response and the trajectory of the stimulus delivery encoded into the ARM. Testing of C57BL/6J (n=15) at baseline found significant

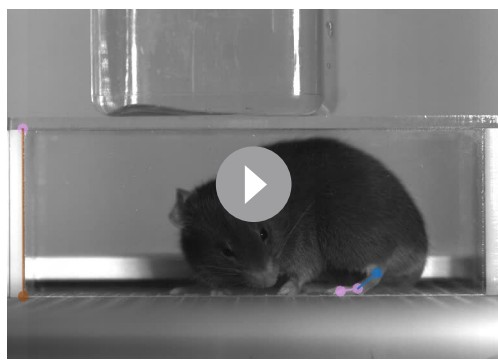

**Video 5.** Representative example of Social LEAP Estimates Animal Poses (SLEAP) tracked 2000 frames per second (fps) pinprick trial, with manual stimulus delivery. Stimulus is delivered at an angle, almost hitting the mouse's paw a second time. Mouse gives a robust response, including guarding behavior.

https://elifesciences.org/articles/99614/figures#video5

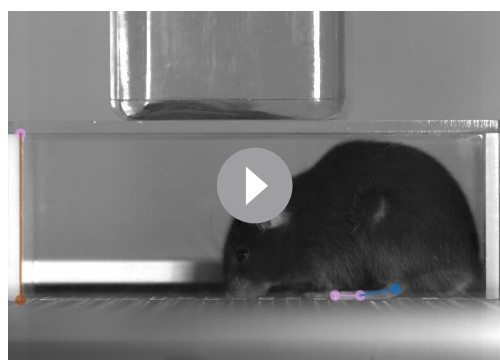

**Video 6.** Representative example of Social LEAP Estimates Animal Poses (SLEAP) tracked 2000 frames per second (fps) pinprick trial, with automated reproducible mechanostimulator (ARM) stimulus delivery. Stimulus is delivered straight up and down, withdrawing quickly after reaching its apex. Mouse gives a robust response, including shaking and guarding behavior.

https://elifesciences.org/articles/99614/figures#video6

decreases in withdrawal latency for pinprick compared to cotton swab stimuli delivered in identical ways by the ARM (*Figure 3—figure supplement 1B*) based on a two-tailed Student's t-test. This decrease was found in both male and female mice, indicating that mechanical withdrawal latency can reliably distinguish between responses to noxious and innoxious mechanical stimuli in rodents.

During the previously described von Frey experiments, it was observed that vFHs never bend back to the exact same position. In addition, stimulus options would be limited by their physical properties. To address this, a more accurate method for measuring withdrawal latency and improving stimulus accuracy, a new force sensor-guided stimulus was developed. This stimulus, either blunt or needle-like, measured the force applied against it in real time and used it to detect paw contact, paw withdrawal, and inform stimulus delivery. The two primary ways it is used are to either apply a force ramp where the force exerted on the mouse paw will increase from 0.5 to 8 g over the course of 5 s (*Figure 3D*) or a consistent force where stim velocity will be adjusted up and down to maintain a consistent force on the paw (*Figure 3E*) for a force and duration of the researcher's choice. Upon paw withdrawal, the stimulus retracts automatically and max force is reached, and withdrawal latency is reported. Through this, we were able to both add an efficient way of measuring mechanical threshold similar to the electronic von Frey, while also giving an option that could be customized based on the needs of the researcher and the specific phenotype of a pain model.

Changes were made to PAWS to make it compatible with frame rates lower than 2000 fps. This was tested using a 0.4 MP, 522 FPS, Sony IMX287 camera recording at 500 fps, and data recorded at 2000 fps by the previously used Photron Fastcam (*Figure 3—figure supplement 1B–E*). The camera paired with PAWS was found to be sufficient to distinguish between cotton swab and pinprick withdrawal responses, suggesting it may be a useful tool for labs that cannot invest in a more expensive device. PAWS features measured from 500 fps video data were not significantly different from the 2000 fps data based on a two-way ANOVA.

To validate the updated PAWS software and reinforce previous findings, a carrageenan inflammatory pain model was used. Mice injected with carrageenan (n=15) showed elevated shaking behavior (p=0.038, two-way ANOVA, post hoc Tukey test) in response to pinprick stimuli in comparison to measurements at baseline (*Figure 3F*). This aligned with previous findings where PAWS has detected elevations in shaking and/or guarding behavior, examples of affective pain behavior, and post-peak paw distance traveled, which correlates with these behaviors in carrageenan pain models and has been found to be a good measure of them in past studies (*Bohic et al., 2023*). To validate the automated mechanical threshold measurements of the new force sensor stimulus, carrageenan and CFA models were used. For the carrageenan model, three replicates of the force ramp stimulus were delivered to each paw, and catch trials were performed every third trial to test whether the mice would respond to the noise of the ARM alone. During catch trials, the stimulus was delivered to the open air behind the mouse, and any movement within 5 s of stimulus delivery was counted as a response. These trials found a 96% response rate in true trials, with only a 7% rate in catch trials, indicating responses were not being driven by device noise (*Figure 3—figure supplement 1F*). The force ramp stimulus successfully detected carrageenan hypersensitivity to touch (*Figure 3G*), with a significant decrease in mechanical threshold from baseline for the ipsilateral paw but not the contralateral control (p=0.027, two-way ANOVA post hoc Tukey test).

As further validation and to test the newly developed consistent force stimulus, 10 mice were first tested at baseline with the 1 g force stim and stim ramp and then tested again 24 hr after CFA

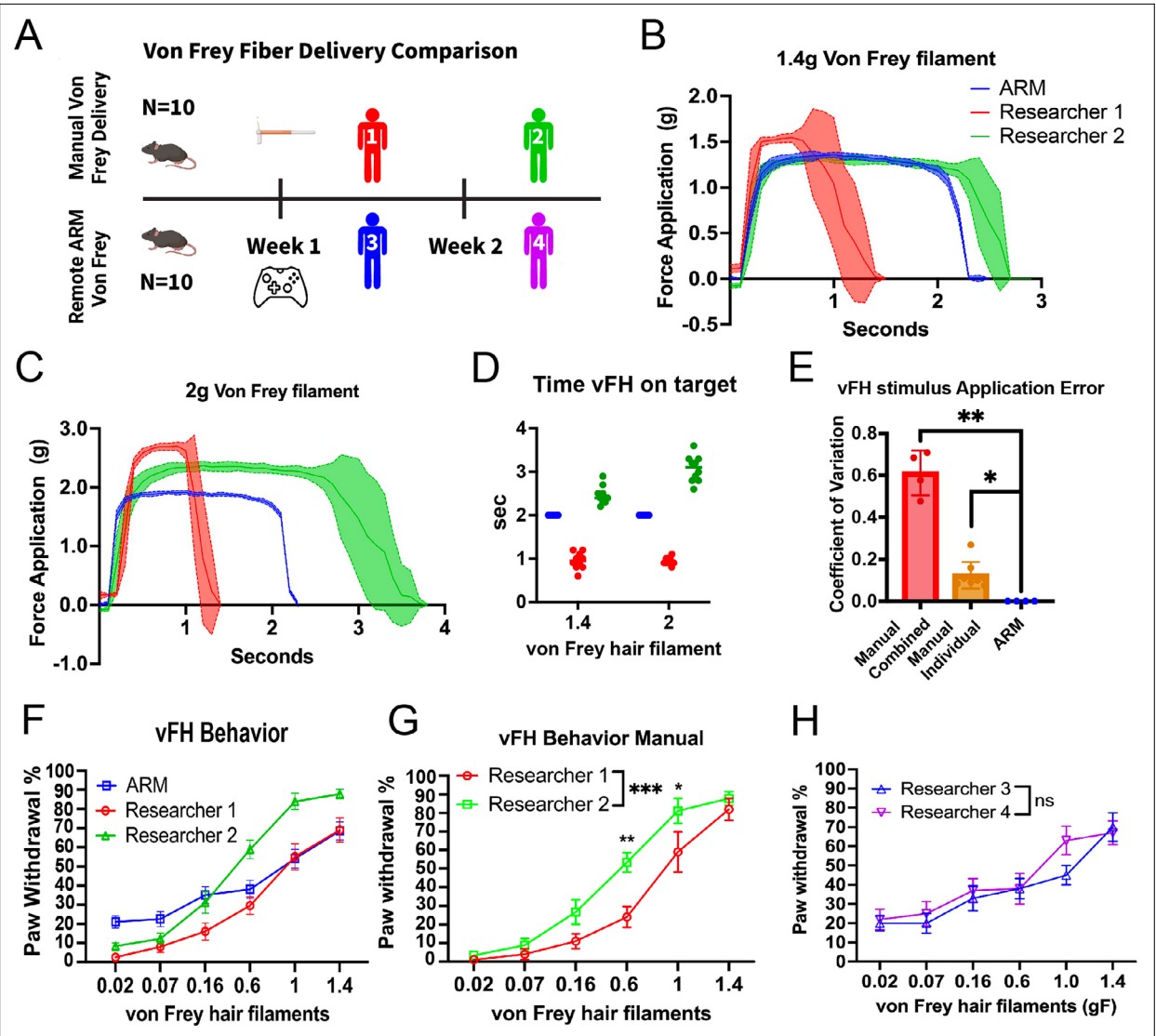

**Figure 2.** The automated reproducible mechanostimulator (ARM) decreases variability in von Frey hair (vFH) stimulus delivery. (**A**) The ARM and external testers each first applied vFH stimulus to a force sensor (1.4 g, 2 g) before applying stimuli to a cohort of mice (n=10) and comparing behavior (0.02 g, 0.07 g, 0.16 g, 0.6 g, 1 g, 1.4 g). (**B**) Researchers and the ARM user were told to apply stimulus for 2 s to the force sensor for 1.4 g (**C**) and 2 g vFHs. (**D**) Stimulus delivery time for 1.4 g and 2 g force sensor trial. (**E**) Coefficient of variance for vFH (0.6 g, 1 g, 1.4 g, 2 g) on target time as determined by the force sensor was calculated for the ARM and compared to each researcher (p=0.0211), and the combined manual trials (p<0.0001) with a one-way ANOVA. (**F**) Both researchers and the ARM tested a cohort of wild-type mice (n=10), applying each vFH 10 times to each mouse, producing the expected vFH response curves, including SEM. (**G**) Comparison between paw withdrawal frequency elicited by Researcher 1 vs Researcher 2 with two-way ANOVA. Significant differences were found in behavior elicited by 0.6 g (p=0.0034), 1 g (p=0.0462), and overall (p=0.0008). (**H**) Two researchers applied ARM vFH stimulus remotely over 2 days. Two-way ANOVA detected no significant differences.

The online version of this article includes the following figure supplement(s) for figure 2:

**Figure supplement 1.** Stimulus application time decreases and rudimentary withdrawal latency measurement.

injection. The force ramp again found significant decreases were found both comparing the baseline to 24 hr for the ipsilateral paw (p=0.026) and ipsilateral to contralateral at the 24 hr time point (p=0.044) (*Figure 3H*). Finally, the 1 g stim found a highly significant decrease in withdrawal latency for the ipsilateral paw (p<0.0001) (*Figure 3I*). Data was analyzed using a two-way ANOVA post hoc Tukey test.

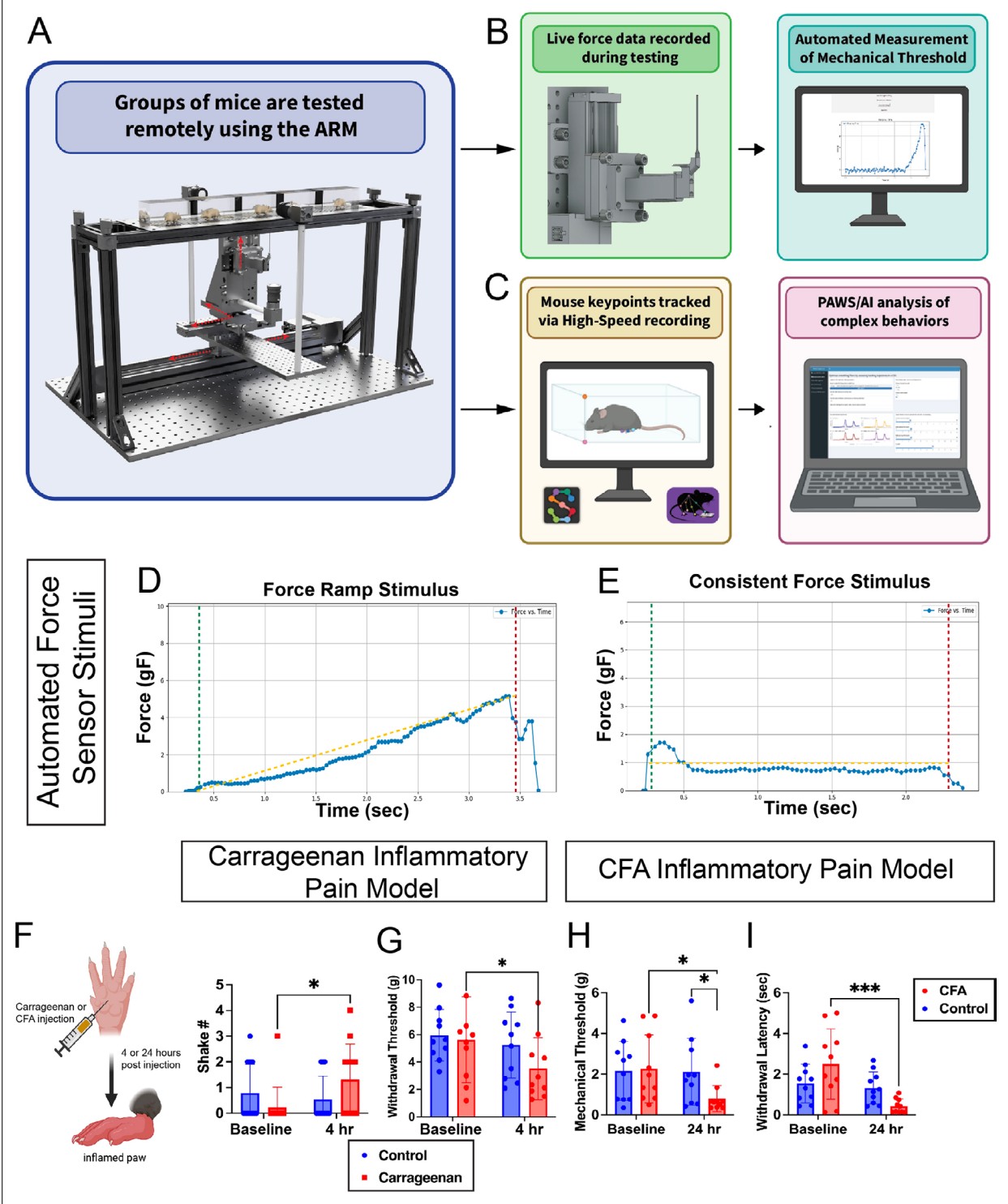

**Figure 3.** Automated reproducible mechanostimulator (ARM) system and integration with pain assessment at withdrawal speeds (PAWS) analysis. (**A**) Groups of mice will be tested using the ARM, (**B**) and for each stimulus delivery, reflexive features including withdrawal latency and mechanical threshold are measured automatically using a force sensor incorporated into the device. (**C**) Pose analysis of the integrated 500 frames per second (fps) cameras and PAWS high-speed analysis then measures the extent of the reaction, including max paw height, velocity, and behaviors associated with the affective aspect of pain, such as distance traveled, shaking, and guarding. (**D**) Integration with load cell allows for customizable force ramp stimulus, where force starts low and ramps up over time, and (**E**) a consistent stimulus that holds at a set force and retracts after duration exceeded or upon paw withdrawal. (**F**) Test of new PAWS pipeline using carrageenan inflammatory pain model detected significantly higher number of paw shakes at 4 hr compared to baseline (p=0.039), and (**G**) decreased mechanical threshold with the initial version of the force ramp stimulus (p=0.027). (**H**) An updated version of the

*Figure 3 continued on next page*

*Figure 3 continued*

force stimulus was used with a CFA model and found significant decreases compared to control and baseline (p=0.044, 0.025) with the ramp stimulus and a highly significant decrease (p<0.0001) in withdrawal latency in response to a 1 g stimulus.

The online version of this article includes the following figure supplement(s) for figure 3:

**Figure supplement 1.** Testing of automated reproducible mechanostimulator (ARM) 500 frames per second (fps) pain assessment at withdrawal speeds (PAWS) analysis and mouse response to catch trials.

## Using the ARM to isolate the effect of the researcher's presence on the mouse withdrawal response

Previous research has found that experimenter sex can have a significant effect on sensitivity to stimulus due to stress-induced analgesia (*Sorge et al., 2014*). It was previously not possible to measure this effect when a researcher was present, but our remote setup of the ARM, though, makes this possible (*Figure 4A*). First, to determine the effect of researcher presence on habituation, two cohorts of male mice (n=10) were habituated for 3 days, 40 min each day with either a researcher present or monitored remotely. Mice were monitored for both the number of times they turned 180° in their chambers and the first point at which they went for a minute resting (no turning, investigating, or grooming). Remote habituated mice showed a significant decrease (p=0.0217, two-way ANOVA) in time to rest over the 3 days (*Figure 4B*), but no significant differences for any single day. The number of turns was measured for each group during the first 10 min of day 1 to act as a baseline, and then from 20 min to 30 min for each day. Turn counts were then compared as a percentage of the baseline count for each group. This period was chosen as it is the period when experiments start after the day of habituation on experimental days. It was found that remote-habituated mice showed significantly less turning on day 2 compared to mice habituated with a researcher present (p=0.024, two-way ANOVA post hoc Tukey test), and that only the remote-habituated mice showed significantly decreased turning behavior on day 3 compared to day 1 (p=0.0234, two-way ANOVA post hoc Tukey test) (*Figure 4C*). These findings indicate that mice take longer to habituate to experimental conditions when an experimenter is present – a result consistent with mice being prey animals on heightened alert for danger.

To determine the effect of the experimenter's presence in isolation, a cohort of wild-type male and female mice was given an innocuous (cotton swab) and then noxious (pinprick) stimuli via remote control of the ARM when either one of two researchers or no researcher was present (*Figure 4D*). Experiments were designed so that circadian rhythm, order of experiments, or day of experiment would not confound the results. Researcher 2 was a male graduate student and Researcher 1 was a female lab technician.

Sex-dependent differences were found in reflexive and affective behavioral components of the mouse withdrawal response when a researcher was present vs not for both reactions to innocuous and noxious stimuli. A two-way ANOVA and a post hoc Tukey test found that cotton swab stimuli elicited increased male mouse reflexive paw withdrawal features, including max paw height (p=0.0413) and max paw velocity (y-axis) (p=0.042) when Researcher 1 was present compared to when no researcher was present (*Figure 4E and F*). Pinprick stimuli (*Figure 4H and I*), on the other hand, led to increased max paw height (p=0.044) and max paw velocity (y-axis) (p=0.041) in male mice compared to female mice when Researcher 1 was present.

Analysis of the shaking behavior elicited by cotton swab and pinprick stimuli found no significant differences in shaking behavior duration (*Figure 4—figure supplement 1*) but found sex-dependent differences in paw distance traveled after the initial withdrawal, including during shaking and guarding behaviors. For cotton swab (*Figure 4G*), male mice showed significantly increased paw distance traveled compared to female mice when Researcher 2 was present (p=0.047, two-way ANOVA post hoc Tukey test) but not when Researcher 2 was present or no researcher was present. Pinprick stimuli also elicited sex-based increases in paw distance traveled (*Figure 4J*) in male mice when Researcher 2 was present compared to both male mice when no researcher was present (p=0.015, two-way ANOVA post hoc Tukey test) and female mice when Researcher 1 was present (p=0.0038, two-way ANOVA post hoc Tukey test).

These results indicate that the researcher's presence at baseline can lead to significant differences in reflexive and affective pain behaviors. In this case, male mice showed increased behavioral responses to both touch and pain behaviors depending on whether the researcher was present. This

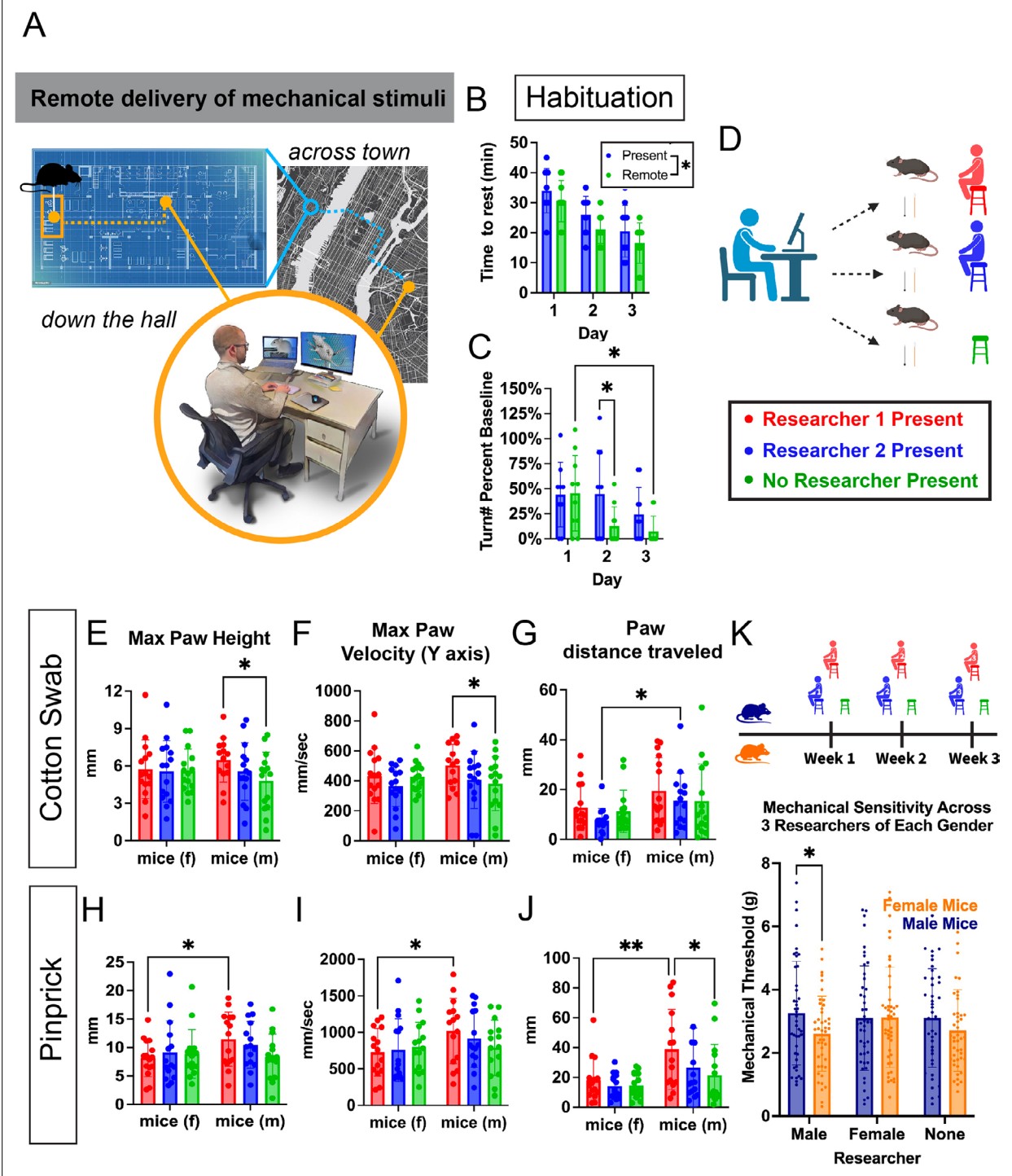

**Figure 4.** Remote delivery of mechanical stimuli reveals the effects of researcher presence. (**A**) Schematic showing the remote operation of the automated reproducible mechanostimulator (ARM) allowing for researcher-agnostic experiments and flexibility. (**B**) Male mice (n=10) were habituated either with a researcher present or not for 3 days. Across the 3 days mice rested for the full minute significantly sooner than those with a researcher present (p=0.0217). (**C**) The number of times each mouse turned, as measured during two 1 min windows 20–30 min each day, normalized by each group's turning behavior during the first 10 min of day 1. On day 2, the remote-habituated mice showed significantly decreased turning behavior compared to those habituated with a researcher present (p=0.024). Only the remote-habituated mice showed significantly decreased turning behavior on day 3 compared to day 1 (p=0.0234). (**D**) Experimental schematic showing remote ARM stimulus delivery with either a researcher or no researcher in the room. (**E–F**) A two-way ANOVA found significant differences in max paw height (p=0.0413) and max y-velocity (p=0.0406) in response to cotton swab for male mice when Researcher 2 was present compared to no researcher. (**G**) Sex-dependent differences were found in response to cotton swab when

*Figure 4 continued on next page*

*Figure 4 continued*

Researcher 1 was present for distance traveled (p=0.0468). (**H–J**) Sex-dependent differences were found in response to pinprick stimuli when Researcher 2 was present, but not other conditions for max paw height (p=0.0436), max y-velocity (p=0.0424), and distance traveled (p=0.0038). Male mice showed significant differences in paw distance traveled (p=0.0149) when Researcher 2 was present compared to when none was. (**K**) To test whether the gender of the experimenter present affects behavior, cohorts of male and female mice (N=15) were tested across 3 weeks with groups of male and female researchers (N=3) and no researcher controls. Male researchers induced significantly decreased mechanical threshold in female vs male mice (p=0.034).

The online version of this article includes the following figure supplement(s) for figure 4:

**Figure supplement 1.** The number of shakes was found to not significantly change based on experimenter presence.

led to sex differences in the affective and reflexive component of the withdrawal response when a researcher is present, which disappears when no researcher is present, or a different researcher is present. For this set of researchers, the female researcher elicited the greater behavioral effect. This appeared at first to contradict previous findings (*Sorge et al., 2014*), but it was hypothesized that the effect of an individual researcher could easily vary compared to their larger gender group. To test this, six new researchers, half male and half female, were recruited and a new cohort of mice (n=15 male, n=15 female) was tested in each of their presence over the course of 3 weeks, controlling for circadian rhythms (*Figure 4K*). The newly added force ramp stimulus type was used for these experiments, with three replicates per trial, to efficiently measure mechanical threshold in a manner comparable to previous work. It was found that female mice showed significantly decreased mechanical threshold compared to male mice (p=0.034, Šídák's multiple comparisons test and Student's t-test) when a male researcher was present. This did not occur when a female researcher or no researcher was present. In the latter case, a slight trend toward this effect was observed, but it was not significant (p=0.21) and may be the result of a single male researcher being responsible for handling and setting up the mice for all experiments.

These findings indicate that sex-dependent differences in evoked pain behavior can appear and disappear based on which researcher/s are in the room. There is a trend toward male researchers overall having a greater effect, but individuals may have a greater or lesser effect on mouse behavior independent of the gender or sex. This could be due to factors such as researcher anxiety, hygiene, pet ownership, and others. This presents a confound that must be considered in the analysis of sex differences in pain and touch behavior, which may explain some of the variation in findings from different researchers. Together, these results suggest that remote stimulus delivery may be the best way to eliminate variation caused by experimenter presence while making it easier to compare with data from researchers in your lab and others.

## Variation in stimulus delivery significantly affects mouse behavioral response

Analysis of high-speed videos of manual pinprick stimulus delivery found substantial variation in the speed, angle, timing, and max height of the stimulus (*Figure 1—figure supplement 1 Videos 3–4*). This variation was apparent between researchers and within tests from a single researcher. It was predicted that stimulus variability would not fully explain the variation in behavioral responses, as other environmental factors – such as experimenter presence and innate variability in animal behavior – also contribute, but would contribute significantly to that variability and could skew data, if biased. To determine the effects of variation in stimulus intensity on mouse withdrawal response, a cohort of male wild-type mice (n=15) was given pinprick stimulus nine times across 3 days, each with a different stimulus apex (1–5 mm above the mesh), but with time to apex and total time above mesh kept consistent (*Figure 5A*). This was done to keep the total potential stimulus exposure time consistent while varying intensity.

Analysis of the resulting withdrawal responses in male mice using PAWS and linear regression found both features with significant positive linear relationships with stimulus intensity and features where no significant correlation could be found. Reflexive features including paw withdrawal latency (*Figure 5B*) and max paw height (*Figure 5C*) showed significant correlations with stimulus intensity that explained 26.9% and 13.8% of the variation in the data, respectively. In contrast, analysis of affective features, including paw shaking duration (*Figure 5D*) and paw distance traveled (*Figure 5E*), found either no significant correlation or a correlation that only explained 6.596% of the variation in

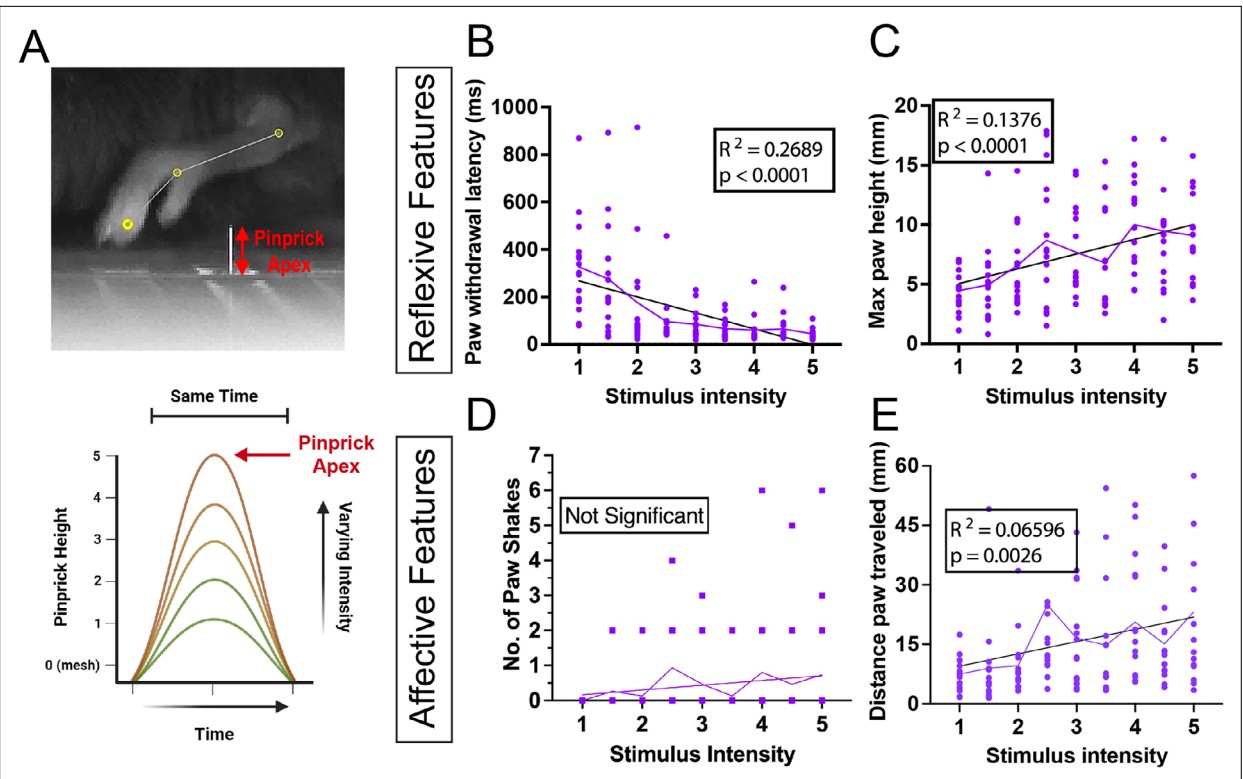

**Figure 5.** Isolating the effect of variation in the application of pinprick stimulus. (**A**) Schematic showing how stimulus delivery variation was modeled through changing pinprick intensity by increasing/decreasing pinprick apex and velocity. (**B–C**) Reflexive features were found to correlate with stimulus intensity based on a simple linear regression, withdrawal latency with a negative correlation, and max paw height with a positive correlation. (**D–E**) For affective features, paw shaking time showed no significant correlation with stimulus intensity, and paw distance traveled showed a positive correlation.

The online version of this article includes the following figure supplement(s) for figure 5:

**Figure supplement 1.** Isolating the effect of variation in applying pinprick stimulus in female mice.

the data. This data indicates that in isolation, stimulus variability has a greater effect on the mouse's initial reflexive response than the following affective features.

A second cohort of female mice was tested to confirm these results (n=15). Reflexive features were again shown to significantly correlate with stimulus intensity, explaining 14.01% of the variability in withdrawal latency (*Figure 5—figure supplement 1A*) based on linear regression. Stimulus intensity significantly correlated with max paw height but only explained 5.22% of the variability based on a simple linear regression. Pair-wise analysis, however, found that a simple linear regression explained 23.02% of the variation in the data from 1 mm to 3 mm with a positive correlation with stimulus intensity and 12.39% from 3 mm to 4.5 mm with a negative correlation with stimulus intensity (*Figure 5—figure supplement 1B*). This may indicate that increasing stimulus intensity at baseline can run into a ceiling effect in terms of its effect on behavioral features. Whether female mice exhibit this effect, but not male mice, due to differences in sex or environmental confounders remains unclear. Analysis of affective pain behavior found no significant correlations between shaking time (*Figure 5—figure supplement 1C*) or paw distance traveled (*Figure 5—figure supplement 1D*).

In summary, variability in stimulus intensity in isolation contributes significantly to the resulting paw withdrawal response, though it appears to primarily affect the initial reflexive response. This is consistent with earlier data, where variation in von Frey delivery time appeared to correlate positively with withdrawal % (*Figure 2*). It should be noted that withdrawal percentage, withdrawal latency, and withdrawal threshold are the most commonly used measures of mechanical sensitivity/pain and are based on the reflexive behavioral response that is significantly correlated with stimulus variability.

## Syncing of stimulus with video and in vivo brain imaging data

Finally, we were interested in testing the performance of the ARM and PAWS analysis of neural activity in a brain region linked to pain. Moreover, we wanted a simultaneous readout of pain behavior with brain activity to confirm that the brain is indeed tracking mechanical stimuli at sub-second resolution. The basolateral amygdala (BLA) was chosen based on previous work that has identified neural populations linked to defensive coping behaviors like paw attending, paw guarding, and paw shaking (*Corder et al., 2019*; *Jones et al., 2020*). This has included both the identification of excitatory neural populations that when activated lead to increased (*Becker et al., 2023*; *Han et al., 2010*) or decreased (*Cai et al., 2018*) pain behavior. Adjustments were made to the ARM's Arduino component to sync the ARM's stimulus delivery with both the high-speed camera and Inscopix platform for cellular-resolved microendoscopy. This allowed for the alignment of stimulus behavior and neural data (*Figure 6A*). Mice injected with AAV9-syn-jGCaMP8f-WPRE virus (jGCaMP8f) targeting the BLA were stimulated with cotton swab and pinprick stimuli. Pinprick stimuli were delivered in two manners, the normal stimuli used previously and a greater intensity stimulus with increased speed and apex referred to as max pinprick. This was done to facilitate a greater range of responses. Each stimulus was delivered 15 times across 6 days for a total of 45 events per mouse (n=2).

BLA video data was processed using the IDEAS platform to correct for motion, identify neurons, and measure $\Delta F/F_o$ (*Figure 6C and D*). Peri-event analysis was used to determine the mean change in cell $\Delta F/F_o$ across the total population (*Figure 6—figure supplement 1A and B*) and to identify neurons that were significantly upregulated or downregulated in response to ARM stimulus events (*Figure 6E and F*). Random time points chosen throughout the testing period were used for a comparison background group (*Figure 6E*). Each of the three stimulus types led to significant up/downregulation of neural activity compared to background (*Figure 6G*). This is consistent with previous work that has identified both neural populations up- and downregulated during pain in the BLA (*Becker et al., 2023*; *Han et al., 2010*). Neurons were registered across consecutive days to identify neurons regulated by mechanical pain (29.3%), touch (10.7%), or both (11.4%) mechanical stimuli (*Figure 6H*).

To determine whether BLA pain neuron regulation correlates with sub-second withdrawal behavioral features, we then analyzed individual touch and pain stimulus events. Cells previously identified by the peri-event analysis as up- or downregulated during mechanical pain were analyzed for each individual event using the Wilcoxon rank-sum analysis to determine the proportion of pain-regulated cells that showed associated up- or downregulation. Cotton swab events showed a significantly smaller proportion of pain matching downregulation, upregulation, or combined group regulation compared to the pinprick stimuli (*Figure 6—figure supplement 1C*). No significant difference was found between pinprick and max pinprick behavior features or pain cell regulation. This may result from a similar ceiling effect to that seen in female mice in the pinprick variation experiments (*Figure 6—figure supplement 1B*). For each event, paw withdrawal latency, max height, max y-velocity, and distance traveled in the 1.5 s following the stimulus were measured using PAWS. These metrics were then plotted against the proportion of upregulated, downregulated, and total BLA mechanical pain-regulated neurons. Paw height and paw velocity were found to be consistent with data from *Figure 4E–I* (male researcher and male mice) and *Figure 5C* (stimulus intensity 2.5 and 4.5) for similar data, with slightly elevated measures of paw distance traveled and decreased paw withdrawal latency for the pinprick stimulus. This was likely caused by sensitization due to multiple stimulus deliveries over the course of the experiment, as logistics required 30 stimulus trials to be delivered per session, compared to the maximum of 3 that were performed during previous experiments.

Our results on withdrawal latency were consistent with previous work (*Corder et al., 2019*) that found that BLA mechanical pain neural activity correlates with this metric, with downregulated (p=0.0032) and total mechanical pain neuron (p=0.0047) proportions correlating with withdrawal latency based on a Pearson correlation, total cell proportion explaining 8.54% of the variation in the data (*Figure 6I*). In comparison, paw distance traveled (*Figure 6J*) and max y-velocity (*Figure 6—figure supplement 1E*) each show correlations with up (p=0.0014, p=0.0324), down (p=0.0062, p=0.0009), and total (p=0.0003, p=0.0007) regulated mechanical pain cells, respectively. Each explained a greater proportion of variability in the data than withdrawal latency, with paw distance traveled explaining 13.88% and max y-velocity explaining 12.11%. Max paw height showed the least correlation with BLA mechanical pain neural activity, showing only correlations with down (p=0.0459) and total (p=0.0426) neural activity, the latter explaining only 4.49% of the variability in the data (*Figure 6—figure supplement*

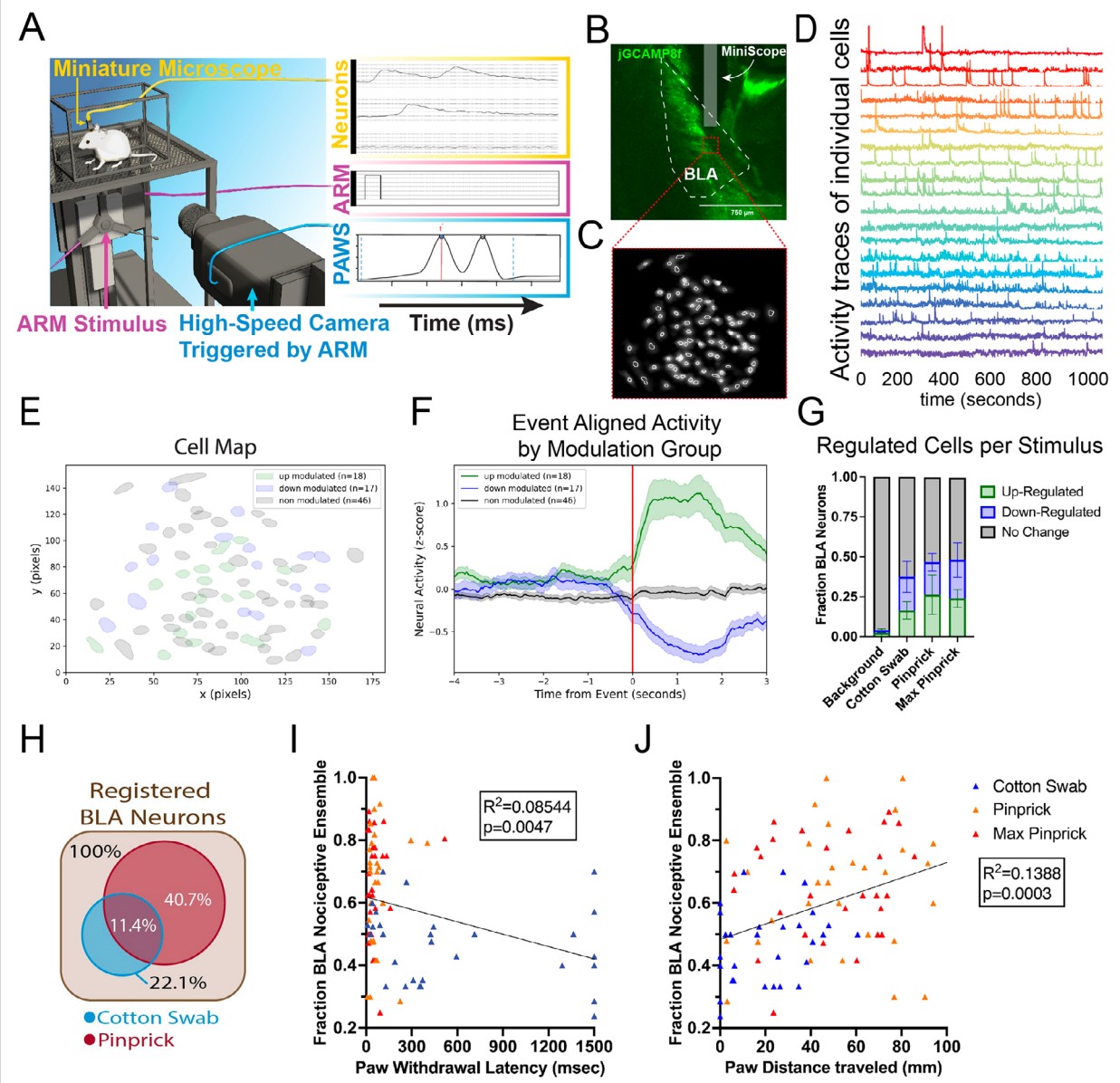

**Figure 6.** Linking automated reproducible mechanostimulator (ARM) stimulation with behavior and cellular-resolved brain activity in the basolateral amygdala (BLA). (**A**) Schematic showing alignment of BLA neural activity recorded by a microendoscope, pain assessment at withdrawal speeds (PAWS) behavioral features, and stimulus facilitated by the ARM. (**B**) Confirmation of injection of jGCaMP8f virus and insertion of Inscopix mini-scope to the BLA. (**C–D**) Cell map from processed mini-scope recording with a selection of representative deconvolved cell traces in pseudocolors over a 1000 s window. (**E–F**) Example traces and cell map of pinprick stimulus aligned up- and downregulated cells based on peri-event analysis. (**G**) Results of peri-event analysis with up- and downregulated cells based on stimulus, and comparison with random background events. Total regulated cells increased compared to background control for all stimuli (p<0.0001). (**H**) Percentage of cells registered across multiple days that are regulated during response to mechanical touch and/or pain stimuli. (**I**) Pearson correlation between the fraction of total peri-event analysis identified mechanical pain-regulated cells with matching regulation for each stimulus event with withdrawal latency (**J**) and distance traveled in the 1.5 s post-stimulus application.

The online version of this article includes the following figure supplement(s) for figure 6:

**Figure supplement 1.** Correlation of additional pain assessment at withdrawal speeds (PAWS) features with basolateral amygdala (BLA) mechanical pain neuron regulation.

*1D*). These findings suggest that max y-velocity and paw distance traveled may be more useful metrics for the study of the BLA's role in pain compared to max paw height or the traditionally used withdrawal latency. These findings may be consistent in other brain regions with neural populations linked to pain, but this remains a subject for future study.

This data indicates that the ARM is an effective tool for efficiently correlating in vivo imaging data with evoked behavioral data, including sub-second behavior. One limitation is that the neural response appears to begin slightly before stimulus impact (*Figure 6F*, *Figure 6—figure supplement 1B*). This was likely caused by a combination of the imprecise nature of ARM v1 paw contact detection and slight delays in the paw contact signal reaching the Inscopix device due to flaws in the software and hardware used, slowing down the signal. Improvements are being made to eliminate this delay as part of the ARM v2.

## Discussion

In this study, we created the ARM, a device for automating mechanosensory assays. The ARM decreased variability in the application of traditional vFH filaments while decreasing the time needed per experiment and eliminating significant variation that was observed between researchers. Using the PAWS pain assessment software, we isolated the effects of experimenter presence and stimulus variability on multiple measures of pain behavior, including paw withdrawal latency. Experimenter presence significantly affected both reflexive and affective measures of paw withdrawal response, leading to the appearance of sex-dependent differences that did not appear when no researcher was present. In contrast, stimulus delivery variability had a greater effect on reflexive measures of the paw withdrawal response compared to affective measures. Based on user data from this work, the device was then redesigned to safeguard vulnerable components from mouse waste, reduce its size to be more in line with traditional von Frey setups, with an option for extension to increase capacity to 10 mice, preparing it for further beta testing outside the Abdus-Saboor lab. To overcome the physical and application limitations of von Frey filaments, a new stimulus incorporating a force sensor was developed and validated to allow for both consistent and customizable flat force application similar to current vFHs and a force ramp similar to eVF devices. Finally, we used the ARM with an Inscopix setup to sync and correlate stimulus, BLA neural activity, and PAWS-measured behavioral features. We identified pain-regulated BLA neurons that were regulated by painful stimuli and were able to correlate their activity with behavioral features of paw withdrawal to mechanical stimuli.

Previous attempts at automating mechanical stimulus delivery, including the electronic von Frey (*Martinov et al., 2013*) and dynamic plantar aesthesiometer (*Nirogi et al., 2012*), have focused on eliminating variability in stimulus delivery. In contrast to the ARM, both of these devices rely upon a researcher being present to aim or deliver the stimulus, can only deliver vFH-like touch stimuli, and only measure withdrawal latency/force threshold. Additionally, progress has been made in automating stimulus assays by creating devices with the goal of delivering precise optogenetic and thermal stimuli to the mouse's hind paw (*Dedek et al., 2023*; *Schorscher-Petcu et al., 2021*). The Prescott lab also incorporated a component into their design to allow for mechanical stimulation, but this piece appears to be limited to a single filament type that can only deliver a force ramp. As a result, these devices and those previously discussed lack customization for delivering distinct modalities of mechanosensation that the ARM allows for. Moreover, in its current form, the automated aiming of some of these devices may not provide the same resolution or reliability of the ARM in targeting defined targets (*Figure 1C*), such as regions of the mouse paw that might be sensitized during chronic pain states. Due to the nature of machine learning pose estimation, substantial work beyond the capacity of a single academic lab, in standardizing the mouse environment and building a robust model based on an extensive and diverse training dataset, will be necessary for automated aiming to match the reliability and flexibility of manual aiming. That said, we believe this work, along with that of the other groups mentioned, has set the groundwork from which a new standard for evoked somatosensory behavior experiments in rodents will be built.

The ARM was designed to mimic the flexibility of manual delivery, capable of delivering poke (pinprick, vFH, cotton swab), static or dynamic brush, and optogenetic stimuli. For many of these stimulus combinations, the researcher does not need to even enter the room to switch between them. In comparison to manual stimulus delivery or delivery that requires a researcher to be present, the ARM is significantly faster. In addition to taking 50% less time to deliver the same vFH test as a researcher doing so manually, it was found that when experiments were being performed remotely using the ARM, without a researcher present, less time appears to be needed for mice to reach a resting state or reduce turning behavior. This could indicate that remote experiments could reduce habituation requirements for experiments. In line with this previous work, the new force sensor stimulus provides

new stimulus options while avoiding over-sensitization and saving time through decreasing the number of trials needed vs vFH and automated mechanical threshold and withdrawal latency measures. Finally, the ARM can be operated using infrared cameras, opening up the possibility of experiments during the mouse dark cycle, which might be more ethologically relevant to study, given as a nocturnal animal it is their peak time of activity.

Mechanical delivery of stimuli to the rodent hind paw by an experimenter and measurement of the resulting paw withdrawal frequency, force threshold, or latency has been a gold standard for measuring nociception and pain for decades (*Dixon, 1980*; *Deuis et al., 2017*). In this paradigm, the experimenter both delivers the stimulus and scores in real time whether the paw moved after stimulation. This assay requires experimenter dexterity and focus, and thus a well-trained researcher. Moreover, because the experimenter performs these assays in real time (stimulus delivery and paw withdrawal measurement), the sub-second speed of the paw withdrawal precludes a thorough description of all the behaviors that occur to a given stimulus. To add behavioral readouts to these rapid paw withdrawals that can aid in pain assessment, we use a pipeline consisting of high-speed videography, automated paw tracking, and custom software to map pain behavioral features (PAWS) (*Jones et al., 2020*; *Bohic et al., 2023*; *Upadhyay et al., 2024*). We have demonstrated that we can detect acute mechanical pain, inflammatory pain, osteoarthritis pain, and neuropathic pain with this pipeline (*Jones et al., 2020*; *Bohic et al., 2023*; *Upadhyay et al., 2024*). Here, we have updated this approach to make it more user-friendly, lower the financial barrier to entry with cheaper, lower frame cameras, and add more readouts to aid in separating out pain states. Moreover, this pain assessment pipeline is fully integrated with the ARM stimulus delivery, which should increase throughput and robustness in performing short-term and longitudinal nociceptive assays. Lastly, although we use a pain assessment pipeline of high-speed videography with automated measures of pain behaviors, the ARM can be used with traditional measurements of pain assessment such as paw withdrawal frequency, latency to withdrawal, or mechanical withdrawal threshold, with the latest version incorporating automated readouts of these measures.

Finally, we combine ARM stimulation with in vivo brain recording in the basolateral amygdala, an area that has been linked to encoding pain affect, unpleasantness, and negative emotion (*Corder et al., 2019*; *Meng et al., 2022*; *Becker et al., 2023*; *Tanimoto et al., 2003*). Although we focus on the amygdala as a proof-of-principle in this study, future studies could use this setup to combine ARM stimulation with behavior mapping and brain recordings in other cortical and subcortical areas implicated in pain (*Meda et al., 2019*; *Chiang et al., 2020*; *Tan and Kuner, 2021*; *Singh et al., 2020*; *Okada et al., 2021*; *Zhou et al., 2023*; *Li and Yang, 2024*; *Chen et al., 2023*). Historically, pain neuroscientists have focused much attention on the peripheral nervous system – the site of nociceptive transduction. The field has made significant progress with this focus, and therapeutic development has centered primarily on blocking pain at its root within the sensory ganglia. That said, there is abundant evidence in both humans and rodents demonstrating the importance of defined circuits in the brain that help to localize the pain, determine pain intensity, and encode the negative emotional states that occur during pain (*François et al., 2017*; *Meda et al., 2019*; *Corder et al., 2019*; *Lee et al., 2021*; *Kragel et al., 2018*; *Apkarian et al., 2005*; *Tracey et al., 2000*).

The ARM democratizes the study of pain by removing the need to have a well-trained researcher spending hours aiming at the rodent paw. This opens the field up to the vast array of scientists who perform in vivo brain recordings to investigate sensory states. Moreover, researchers outside the field – such as those studying autism, neurodegeneration, or social isolation, all of which have reported somatosensory deficits (*Orefice et al., 2016*; *O'Leary et al., 2018*; *Crane et al., 2009*; *Hu et al., 2023*; *Horiguchi et al., 2013*) – might have an easier time phenotyping their animals with the ARM. It can also not be ignored that traditional somatosensory assays are physically taxing and are not the options for some researchers with physical disabilities – challenges the ARM overcomes in many ways. Opening the pain and somatosensory field should accelerate the pace of discovery.

In conclusion, we have built a device that can deliver a variety of mechanical stimuli, even remotely, at above expert level. We envision the ARM being used across academia and industry to uncover new mechanisms of pain neurobiology and for high-throughput screening of novel analgesics. To promote the widespread adoption of this device across as many labs as possible, a company named Tactorum Inc has been formed.

## Methods

### Mice

All experimental testing was performed in compliance with the Guide for the Care and Use of Laboratory Animals (NIH), with approved IACUC protocol number AC-AABL6560. All procedures were approved by the Institutional Animal Care and Use Committee of Columbia University. Unless stated otherwise, all mice were co-housed with a maximum of four other mice in a large housing room with approximately 100 other mouse cages. C57BL/6J mice were ordered from Jackson Laboratories. Over the course of the experiments, male and female mice ranging from 8 to 16 weeks in age were used for testing. All groups compared were within a week in age of each other. The mice were kept on a day-night light-dark cycle and brought to a specialized behavior analysis room for testing. Mice were normally fed commercially available pelleted rodent chow and watered ad libitum.

### Somatosensory behavior assays

During testing, mice were placed in acrylic chambers (4.2 cm × 11.5 cm × 4.6 cm) on a specialized mesh table held down by acrylic weights in an isolated testing room separate from normal housing. A maximum of five mice was tested at any one time. Mice were allowed to acclimate to their housing for 2 weeks before testing. Before somatosensory testing, mice were habituated for 4 days, 1 hr each, to testing conditions. A habituation program, where the ARM moved randomly and gave stimulus to empty air, was used to acclimate the mice to its noise. For experiments where only remote ARM work would be performed, only 1 day of habituation was found to be needed. On the day of testing, mice were habituated to their chamber for 15 min before testing. During testing, the ARM and high-speed camera moved between fixed starting positions for each chamber with the z-axis at a default working height of 156.25 with the mesh 14 mm above the stimulus. This movement, along with precise movement of the ARM and stimulus delivery, was performed using an Xbox One controller and custom Python code. The bottom-aiming camera was calibrated either by poking a pinprick through a piece of tape and moving its crosshairs to that point or using previously used coordinates. Once calibrated, the bottom camera was used to aim the stimulus at the center of the mouse paw, before delivering stimuli. Cotton swab and pinprick stimuli were delivered using a sine wave motion of the ARM's z-axis starting from the trough with amplitude of 8 mm and wavelength of 0.8 s. For vFH testing, the z-axis started at a working height of 145 mm with mesh 0.14 mm above stimulus and delivered stimulus using a sine wave motion of the ARM's z-axis with an amplitude of 3.5 mm and wavelength of 2.2 s. These values were chosen to model the average manual delivery of stimuli as seen in *Jones et al., 2020*, while avoiding accidental stimulus delivery to body parts other than the paw, or double stimulus of the paw. The radial axis was used to switch between cotton swab, pinprick, and dynamic brush stimulus; it was also used to switch between vFH. Unless otherwise stated, mice were tested remotely with the researcher controlling the ARM from elsewhere in the lab. Stimulus delivery triggered camera recording with a calibrated delay to ensure recordings would start 25 ms before the stimulus went above the mesh to facilitate the measurements of withdrawal latency.

The ARM v2 uses a force sensor-aided stimulus that allows for automated detection of paw contact and withdrawal. Stimulus can be delivered with a fixed force, where the device will depress the paw until a max force of the researchers' choice (0.25–12 g) is reached, maintaining that force until the desired duration is exceeded or withdrawal occurs. Stimulus can also be delivered using a force ramp where the stimulus makes contact with the paw and then increases linearly from 0.5 g to 8 g over the course of 5 s. The ARM system uses custom software reports and keeps track of withdrawal latency and max force applied across all trials. This was used for the CFA experiments; the carrageenan experiments were performed with a prototype version that detected withdrawal latency and max force but delivered stimulus at a consistent velocity before retracting. Three 500 fps cameras (0.4 MP, 522 FPS, Sony IMX287) can also record videos of the mouse behavioral response from both sides and below.

The ARM is the intellectual property of Columbia University, so we are limited in the extent to which we can share building instructions or other such details. The ARM operation software is subject to similar withholding requirements. That said, the current version of the device is licensed by and made available by Tactorum Inc.

For vFH experiments, bioseb von Frey filaments were used either delivering stimulus in the canonical manner (*Dixon, 1980*; *Zhou et al., 2018*) or attached to a holder on the ARM and depressed against the mouse paw in the manner discussed. Testers or the ARM first delivered vFH stimulus

(1.4 g, 2 g) to a force sensor with 0.05 gF resolution (Mark-10 Model M5-05 max 250 GF), before delivering vFH (0.02 g, 0.07 g, 0.16 g, 0.6 g, 1 g, 1.4 g) to mice (n=10 male). Separate cohorts of 10 mice were used for ARM and manual delivery, with a week given between each researcher to avoid sensitization. Each vFH was delivered 10 times consecutively to each mouse and withdrawal frequency was measured. For habituation experiments, 2 groups of 10 male mice were either habituated with a researcher present or without. Mice were habituated five at a time for 3 days 40 min each day with timing and experimenters kept consistent. Mice were monitored remotely in 1 min periods, with 4 min in between as other mice were monitored. Mice were monitored for number of 180° turns and whether they rested (not turning, grooming, or investigating) for the whole minute. For remote delivery experiments, three groups of 5 mice (n=15 male and female) were used with each group either having Researcher 1, Researcher 2, or no researcher present during experiments. This was repeated for 2 more days to ensure each group experienced each condition. For the stimulus variation experiments, nine stimulus types were devised using standard pinprick stimulus as a basis and calculating new sine waves to vary pinprick apex from 1 mm to 5 mm in 0.5 mm steps while keeping the time the pinprick spent above the mesh consistent. Mice (n=15 male and female) were then delivered a random selection of these stimuli, 3 per day for 3 days, with none repeated so that each mouse would by the end receive all 9.

### ARM targeting experiment

Five researchers delivered pinprick stimuli to a target, 10 times manually and 10 times with the ARM. Stationary 0.5 mm diameter dots on printer paper were used as the target for these experiments. 20 targets were used per researcher, 10 for manual and 10 for ARM. Researchers were instructed to aim for the center of each dot and deliver stimulus poking through the paper. Calipers were then used to measure the distance from each hole or indentation to the center of the corresponding target.

### Carrageenan inflammatory pain assay

Mice were first tested with cotton swab and pinprick stimuli by the ARM. Mice were then injected with 20 µl 3% $\lambda$-Carrageenan (Sigma-Aldrich) in 0.9% sterile NaCl solution (saline) into the hind paw. 4 hr post-injection, they were again tested with cotton swab and pinprick stimuli.

### Analysis of paw withdrawal behavior

We utilized PAWS as a comprehensive behavioral tool to assess the reflexive and affective components of the evoked paw withdrawal response as previously described (*Jones et al., 2020*). The reflexive component describes the initial rapid paw withdrawal, putatively governed by the peripheral nervous system and spinal cord, while the affective component describes the rest of this response, putatively governed by the brain. PAWS distinguishes the reflexive from the affective portions of the response (designated as t*), which is the time point in the response at which the paw reaches its first local maximum in height. PAWS analyzes these components separately and extracts kinematic features such as maximum height, maximum x-velocity, maximum y-velocity, distance traveled in both the reflexive and affective domains. For this paper, max paw height and max y-velocity were extracted from the reflexive domain, and distance traveled was extracted from the affective domain. Within the affective metrics, PAWS additionally extracts number of shakes (defined as a rapid velocity inflection), total duration of shaking behavior, and total duration of guarding behavior (defined as elevation of the paw above a specified height).

We recorded evoked paw withdrawal responses to cotton swab, dynamic brush, and light and heavy pinprick mechanical stimuli using a high-speed video camera (Photron FastCAM Mini AX 50 170K-M-32GB - Monochrome 170K with 32 GB memory) and attached lens (Zeiss 2/100M ZF.2-mount) or a lower fps camera (0.4 MP, 522 FPS, Sony IMX287 camera). Videos were recorded at 2000 fps or 500 fps. These videos were saved directly to an external hard drive as .avi or .mp4 format on a Dell laptop with Photron FastCAM Analysis software installed.

We used SLEAP, a machine-learning framework for supervised behavioral tracking of the mouse hind paw (*Pereira et al., 2022*). In training our models, we annotated the heel (labeled 'heel'), the metatarsophalangeal joint (labeled 'center'), and the toe (labeled 'toe'), as well as two reference points set to the top left and bottom left corner of the transparent acrylic chamber housing the mouse during stimulation (labeled 'objectA' and 'objectB,' respectively). The 'center' point was the default

point used for analysis. These reference points were used to automatically scale each video from pixel distances to millimeter distances given a known distance between these points when loaded into PAWS. After training and running inference on unlabeled frames, we exported all tracking data as HDF5 files before PAWS analysis. The machine used to train the SLEAP model was running Windows 11 Pro, an NVIDIA GeForce RTX 3060 GPU, and an Intel Core i7-12700K CPU processor.

We utilized a custom script within PAWS to extract tracking data and tracking confidence scores from HDF5 files into CSVs. For PAWS analysis parameters, we used a built-in average rolling filter with a window size of 17 frames as our default for analysis. We used a p-cutoff threshold of 0.45, at which tracking values below 45% confidence would be replaced with linear interpolation, a shaking height threshold of 0.35 mm, a fixed baseline height of 1 mm, and a y threshold (defining paw lifts) of 0.5 mm. These values were varied when a tracked video could not be analyzed at the default settings. In the subset of videos where we calculated paw withdrawal latency, we fit a sinusoidal stimulus trajectory to the parameters used to deliver pinprick or cotton swab by the ARM. We then flexibly defined withdrawal latency as the point in time following stimulus application at which the tracking data for the body part of interest (heel, metatarsophalangeal joint, or toe) is higher than the stimulus trajectory. Following batch processing of tracked videos, PAWS exports a single CSV spreadsheet containing these individual metrics. We updated PAWS to flexibly scale behavioral tracking data from cameras recorded at less than 2000 fps by defining a custom 'resize' function which expanded the data to its 2000 fps-equivalent size (for instance, 50 data points collected over 0.1 s at 500 fps were expanded to 200 data points, equivalently collected over 0.1 s at 2000 fps), using linear interpolation to estimate the positions of the paw between each point. This resize function can also be utilized for recordings taken over 2000 fps, where instead of interpolation, the trajectories were simply downsampled to 2000 fps. These adjusted trajectories were then processed through our PAWS pipeline.

Our PAWS pipeline is freely available for installation and use on GitHub (https://github.com/osimon81/PAWS, copy archived at *Ogundare, 2023*). For ease of use, we have also developed a comprehensive tutorial with example tracking data and function documentation available through GitHub pages (https://osimon81.github.io/PAWS).

## Stereotaxic surgery

Eight-week-old male C57BL/6J mice were injected with AAV9-syn-jGCaMP8f-WPRE virus (Addgene #162376-AAV9) (jGCaMP8f) and implanted with integrated 0.6 mm × 7.3 mm lens with attached baseplates (Inscopix cat. #1050-004413) via stereotaxic surgery in a single-step procedure. Viruses were injected and lenses implanted at the following coordinates to target the BLA: (AP: –1.6 mm, ML: 3.2 mm, DV: –4.5 mm). All lenses were implanted on the right hemisphere, following the use of a 22G guide needle to clear tissue for the lens down to DV: –4.4 mm. The integrated lenses with baseplates were secured to the skull with Metabond adhesive cement (C&B #S380). Mice were treated with meloxicam for 3 days post-surgery, and the virus was allowed to express for 4 weeks before imaging.

## Microendoscope imaging

Mice were habituated with the dummy microendoscope on the ARM platform for 1 hr the day before the experiment. On each experimental day, mice were scruffed and attached to the mini-epifluorescence microscope via the head-mount cemented onto the skull during surgery. Mice were then habituated on the ARM platform for 5 min, and then 10 min of baseline brain activity was recorded. After baseline was taken, the mouse's left hind paw was given a stimulus every 2 min until 10 successful stimulations had been delivered or until 50 min of total recording time had elapsed. On days 1, 2, and 3 of the experiment, mice were stimulated with cotton swab, dynamic brush, and a light pinprick, respectively. Only one type of stimulus was given per day, and no day of recording exceeded 50 min. Calcium imaging data was collected using the Inscopix nVista system (Inscopix Data Acquisition Software [IDAS Version 1.3.0]). Recordings were taken under the conditions: frame rate = 20–25 Hz, LED power = 0.5 mW/mm$^2$, and a gain and focus that optimized image resolution and GCaMP expression for each mouse. A general-purpose input/output (GPIO) was configured such that triggering the ARM placed an annotation in the Inscopix output. Videos were automatically spatially downsampled by 4 by the data acquisition software, as recommended by the manufacturer.

## Microendoscope imaging fluorescence analysis

Video and annotation files generated during data collection by the Inscopix Data Acquisition Software were uploaded and processed in the Inscopix Data Exploration, Analysis, and Sharing (IDEAS) platform. Videos were motion-corrected with the…and normalized with (each function). Image segmentation and cell detection were performed with the (which pipeline). The Peri-Event Analysis Workflow (Version 2.4.3) was used to define events.

## Imaging statistics and data analysis

Microendoscope data was analyzed using the IDEAS platform for motion correction, application of a spatial bandpass filter, and a constrained non-negative matrix factorization. The resulting cells were then manually accepted or rejected and registered using Inscopix data processing. A peri-event analysis was performed using IDEAS for each recording based on either Inscopix GPIO data from ARM stimulus events or random timestamps used to represent background fluctuation. The statistical windows were –2 to 0 and 0 to 2. Cells with significant regulation during pinprick or max pinprick events and matching registered cells were identified as BLA mechanical pain neurons, which were then analyzed on an individual event basis. A window –4 s to –2 s before each event was used to calculate z-scores weights, and then z-scores from the –4 to –2 window and 0 to 2 window were compared using a Wilcoxon rank-sum test to determine whether significant upregulation or downregulation occurred. Fractions of upregulation and downregulation that matched average mechanical pinprick regulation determined by the peri-event analysis were determined for each event and correlated with max paw height, max y-velocity, withdrawal latency, and distance traveled as measured by PAWS using a simple linear regression and Pearson correlation.

## Acknowledgements

We thank the Advanced Instrumentation Core at Columbia University's Zuckerman Mind Brain Behavior Institute, and in particular Jake Nazarian and Tanya Tabachnik, for their assistance in designing and building the ARM. We thank members of the Abdus-Saboor lab, as well as Jeffrey Mogil, Biafra Ahanonu, Nathan Fried, and the SENse lab at UC Berkeley for helpful discussion and comments on this study and manuscript. We thank members of Justin's thesis committee, Oliver Hobert and Maria Tosces, for helpful suggestions. We acknowledge Janet Sinn-Hanlon for the illustrations. IA-S and lab members acknowledge support from Columbia University start-up funds, Howard Hughes Medical Institute, National Institute of Health New Innovator Award, Rita Allen Foundation, Pew Charitable Trust, Brain Research Foundation, McKnight Foundation, Burroughs Wellcome Fund, Simons Foundation, Alfred P Sloan Foundation, and Chan Zuckerberg Initiative. VEA acknowledges support from the Rita Allen Foundation, Pew Charitable Trust, Alfred P Sloan Foundation, and NIH (R01NS124799, R01NS119268). JB acknowledges support from NIH (F31NS134275) and (5T32DA055569-03).

## Additional information

### Competing interests

Justin Burdge, Ishmail Abdus-Saboor: co-founder of Tactorum, the company that sells the ARM device. Victoria E Abraira: advisor of Tactorum, the company that sells the ARM device. The other authors declare that no competing interests exist.

### Funding

| Funder | Grant reference number | Author |
| --- | --- | --- |
| National Institutes of Health | | Victoria E Abraira<br>Ishmail Abdus-Saboor |
| Howard Hughes Medical Institute | | Ishmail Abdus-Saboor |
| Pew Charitable Trusts | | Ishmail Abdus-Saboor |

| Funder | Grant reference number | Author |
|---|---|---|
| Chan Zuckerberg Initiative | | Ishmail Abdus-Saboor |
| Rita Allen Foundation | | Ishmail Abdus-Saboor |
| Alfred P. Sloan Foundation | | Ishmail Abdus-Saboor |
| National Institutes of Health | R01NS124799 | Victoria E Abraira |
| National Institutes of Health | R01NS119268 | Victoria E Abraira |
| National Institutes of Health | F31NS134275 | Justin Burdge |
| National Institutes of Health | 5T32DA055569-03 | Justin Burdge |

The funders had no role in study design, data collection and interpretation, or the decision to submit the work for publication.

## Author contributions

Justin Burdge, Conceptualization, Data curation, Formal analysis, Validation, Investigation, Visualization, Methodology, Writing – original draft, Writing – review and editing; Anissa Jhumka, David J Margolis, Conceptualization; Ashar Khan, Brittany Bistis, Miao Li, Yosuke M Morizawa, Leah Yadessa, Arlene J George, Abednego Delinois, Wadzanayi Mayiseni, Noah Loran, Data curation; Simon Ogundare, Data curation, Software, Formal analysis, Validation, Visualization; Nicholas Baer, Conceptualization, Data curation, Software; Sasha Fulton, Data curation, Formal analysis, Visualization; Alexander Kaplan, Andre Toussaint, Data curation, Formal analysis; William Foster, Conceptualization, Data curation, Software, Formal analysis; Joshua Thackray, Software; Jake Nazarian, Guang Yang, Conceptualization, Resources; Victoria E Abraira, Conceptualization, Supervision, Writing – review and editing; Ishmail Abdus-Saboor, Conceptualization, Resources, Supervision, Funding acquisition, Visualization, Writing – original draft, Project administration, Writing – review and editing

## Author ORCIDs

Alexander Kaplan ⓘD https://orcid.org/0000-0002-4634-4322
Joshua Thackray ⓘD https://orcid.org/0000-0003-2828-452X
Abednego Delinois ⓘD https://orcid.org/0009-0007-9071-8516
Guang Yang ⓘD https://orcid.org/0000-0002-5739-9126
Ishmail Abdus-Saboor ⓘD https://orcid.org/0000-0003-2120-0063

## Ethics

All experimental testing was performed in compliance with the Guide for the Care and Use of Laboratory Animals (NIH). All procedures were approved by the Institutional Animal Care and Use Committee of Columbia University. Unless stated otherwise all mice were co-housed with a max of 4 other mice in a large housing room with approximately 100 other mouse cages. C57BL/6J mice were ordered from Jackson Laboratories. Over the course of the experiments, male and female mice ranging from 8-16 weeks in age were used for testing. All groups compared were within a week in age of each other. The mice were kept on a day-night light-dark cycle and brought to a specialized behavior analysis room for testing. Mice were normally fed commercially available pelleted rodent chow and watered ad libitum.

Reviewer #1 (Public review): https://doi.org/10.7554/eLife.99614.3.sa1
Reviewer #2 (Public review): https://doi.org/10.7554/eLife.99614.3.sa2
Reviewer #3 (Public review): https://doi.org/10.7554/eLife.99614.3.sa3
Author response https://doi.org/10.7554/eLife.99614.3.sa4

# Additional files

## Supplementary files

MDAR checklist

## Data availability

All original data for this paper can be freely downloaded from Dryad here: https://doi.org/10.5061/dryad.cnp5hqcj7. The ARM is the intellectual property of Columbia University, so we are limited in the extent to which we can share building instructions or other such details. The ARM operation software is subject to similar withholding requirements. That said, the current version of the device is licensed by and made available by Tactorum Inc. Further information and requests for resources and reagents should be directed to and will be fulfilled by lead contact, Dr. Ishmail Abdus-Saboor ( ia2458@columbia.edu).

The following dataset was generated:

| Author(s) | Year | Dataset title | Dataset URL | Database and Identifier |
| --- | --- | --- | --- | --- |
| Abdus-Saboor I, Burdge J | 2025 | Remote automated delivery of mechanical stimuli coupled to brain recordings in behaving mice | https://doi.org/10.5061/dryad.cnp5hqcj7 | Dryad Digital Repository, 10.5061/dryad.cnp5hqcj7 |

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
