## [Editor Report · eLife Assessment]

This **important** study describes the development and validation of an Automated Reproducible Mechano-stimulator (ARM), a tool for standardizing and automating tactile behavior experiments. The data supporting the use of the ARM system are **compelling**, and demonstrate that by removing experimenter effects on animals, it reduces variability in various parameters of stimulus application. Moreover, the authors demonstrate that any noise emitted from the ARM does not induce an increased stress state. Once commercially available, the ARM system has the potential to increase experimental reproducibility between laboratories in the somatosentation and pain fields.

---

## [Referee Report · Reviewer #1 (Public review)]

Allodynia is commonly measured in the pain field using von Frey filaments, which are applied to a body region (usually hindpaw if studying rodents) by a human. While humans perceive themselves as being objective, as the authors noted, humans are far from consistent when applying these filaments. Not to mention, odors from humans, including of different sexes, can influence animal behavior. There is thus a major unmet need for a way to automate this tedious von Frey testing process, and to remove humans from the experiment. I have no major scientific concerns with the study, as the authors did an outstanding job of comparing this automated system to human experimenters in a rigorous and quantitative manner. They even demonstrated that their automated system can be used in conjunction with in vivo imaging techniques.

While it is somewhat unclear how easy and inexpensive this device will be, I anticipate everyone in the pain field will be clamoring to get their hands on a system like this. And given the mechanical nature of the device, and propensity for mice to urinate on things, I also wonder how frequently the device breaks/needs to be repaired. Perhaps some details regarding cost and reliability of the device would be helpful to include, as these are the two things that could make researchers hesitant to adopt immediately.

The only major technical concern, which is easy to address, is whether the device generates ultrasounic sounds that rodents can hear when idle or operational, across the ultrasonic frequencies that are of biological relevance (20-110 kHz). These sounds are generally alarm vocalizations and can create stress in animals, and/or serve as cues of an impending stimulus (if indeed they are produced by the device).

Comments on revisions:

Was Fig. 1 updated with the new apparatus design? i.e. to address issue of animal waste affecting function over time?

I have no further comments.

---

## [Referee Report · Reviewer #2 (Public review)]

Summary:

Burdge, Juhmka et al describe the development and validation of a new automated system for applying plantar stimuli in rodent somatosensory behavior tasks. This platform allows the users to run behavior experiments remotely, removing experimenter effects on animals and reducing variability in manual application of stimuli. The system integrates well with other automated analysis programs that the lab has developed, providing a complete package for standardizing behavior data collection and analysis. The authors present extensive validations of the system against manual stimulus application. Proof of concept studies also show how the system can be used to better understand the effect of experimenters on behavior and the effects of how stimuli are presented on the micro features of the animal withdrawal response.

Strengths:

If widely adopted, ARM has the potential to reduce variability in plantar behavior studies across and within labs and provide a means to standardize results. It provides a way to circumvent the confounds that humans bring into performing sensitive plantar behavior tests (e.g. experimenter odors, experince, physical abilities, variation in stimulus application, sex). Furthermore, it can be integrated with other automated platforms, allowing for quicker analysis and potentially automated stimulus delivery. The manuscript also presents some compelling evidence on the effects of stimulus application time and height on withdrawals, which can potentially help labs that are manually applying stimuli standardize applications. The system is well validated and the results are clear and convincingly presented. Claims are well supported by experimental evidence.

Weaknesses:

ARM seems like a fantastic system that could be widely adopted, a primary weakness is that it is not currently available to other labs. This will eventually be remedied as it is commercialised.

---

## [Referee Report · Reviewer #3 (Public review)]

Summary:

This report describes the development and initial applications of the ARM (Automated Reproducible Mechano-stimulator), a programmable tool that delivers various mechanical stimuli to a select target (most frequently, a rodent hindpaw). Comparisons to traditional testing methods (e.g., experimenter application of stimuli) reveal that the ARM reduces variability in the anatomical targeting, height, velocity, and total time of stimulus application. Given that the ARM can be controlled remotely, this device was also used to assess effects of experimenter presence on reflexive responses to mechanical stimulation. Although not every experimenter had notable sex-dependent effects on animal behavior, use of the ARM never had this effect (for obvious reasons!). Lastly, the ARM was used to stimulate rodent hindpaws while measuring neuronal activity in the basolateral nucleus of the amygdala (BLA), a brain region that is associated with the negative affect of pain. This device, and similar automated devices, will undoubtedly reduce experimenter-related variability in reflexive mechanical behavior tests; this may increase experimental reproducibility between laboratories who are able to invest in this type of technology.

Strengths:

Clear examples of variability in experimenter stimulus application are provided and then contrasted with uniform stimulus application that is inherent to the ARM.

The ARM is able to quickly oscillate between delivery of various mechanical stimuli; this is advantageous for experimental efficiency.

New additions to the ARM and PAWS platforms have been methodically tested to ensure reproducibility and reliability.

---

## [Author Response]

The following is the authors’ response to the original reviews.

**Reviewer #1 (Public review):**
(1) Given the mechanical nature of the device and the propensity for mice to urinate on things, I also wonder how frequently the device breaks/needs to be repaired. Perhaps some details regarding the cost and reliability of the device would be helpful to include, as these are the two things that could make researchers hesitant to adopt immediately.

We thank the reviewer for their astute observations. We also noted the problem of mouse waste and incorporated this concern into the redesign we mention in the text.

“Mouse waste getting on mechanical parts was found to be a major concern for the initial version of the device. As part of the redesign, the linear stages were moved out from under the mice to avoid this problem. Despite this problem, the original version of the device has not had any of its stages break down yet. A common problem though was that stimulus tips would blunt or break if they hit the mesh of the mesh table, requiring replacement. This has been solved in the latest version through a new feature where the mesh is detected via the force sensor, prompting immediate stimulus withdrawal, avoiding damage.”

In regards to cost and adoption, we have added this sentence to the final line of the discussion:

“To promote wide adaptation of this device across as many labs as possible, a company, Tactorum Inc, has been formed.”

(2) The only major technical concern, which is easy to address, is whether the device generates ultrasonic sounds that rodents can hear when idle or operational, across the ultrasonic frequencies that are of biological relevance (20-110 kHz). These sounds are generally alarm vocalizations and can create stress in animals, and/or serve as cues of an impending stimulus (if indeed they are produced by the device).

The reviewer brings up an interesting question. The ARM does not make a lot of noise, but some of the noise it emits does range into the 20-110 kHz range, though besides this does not qualitatively have other similarities to a mouse vocalization. Based on this we tested whether the noise produced by the ARM causes stress in naïve mice.

“A concern was raised that the noise of the ARM may cause stress in the mice tested. To test this, the open field test was performed with naïve mice (n=10) 2 feet from the ARM while the ARM either sat silent or ran through its habituation program, producing noise. The mouse's center point movement was then tracked in relation to the chamber, its edges, and center. No significant differences were found in distance traveled, center entrances, center, time in center, and latency to center entrance based on a student’s two-tailed t-test (Figure S1D-G). Based on this, neither stress nor locomotion differences were detected by this test, indicating the ARM does not induce an increased stress state due to its noise, even in non-habituated mice.”

(3) This sentence in the intro may be inaccurate: "or the recent emergence of a therapeutic targeting voltage-gated sodium channels, that block pain in both rodents and humans such as VX-548 for NaV1.8 (Jones 2023)" Despite extensive searching, I have been unable to find a reference showing that VX-548 is antinociceptive in rodents (rats or mice). As for why this is the case, I do not know. One speculation: this drug may be selective for the human Nav1.8 channel (but again, I have found no references comparing specificity on human vs rodent Nav1.8 channels). To not mislead the field into thinking VX-548 works for rodents and humans, please remove "both rodents and" from the sentence above (unless you find a reference supporting VX-548 as being effective in pain assays with rodents. There is a PK/PD paper with rodents, but that only looks at drug metabolism, not efficacy with pain assays).

We agree with the reviewer and have removed mention of the new Nav1.8 therapeutic also working in rodents.

(4) In the intro paragraph where variability in measuring mechanical stimuli is described, there is a new reference from the Stucky lab that further supports the need for an automated way to measure allodynia, as they also found variability between experimenters. This would be a relevant reference to include: Rodriguez Garcia (2024) PMID: 38314814.

Thanks to the reviewer for this relevant citation and we have updated the text to incorporate this:

“Recent studies utilizing the manual highspeed analysis of withdrawal behavior analysis developed by Abdus-Saboor et al. 2019 has reproduced this sizable experimenter effect using the new technique. (Rodríguez García 2024)”

(5) "a simple sin wave motion": should be "sine", correct throughout (multiple instances of "sin")

Corrections made where relevant.

**Reviewer #2 (Public review):**
(1) ARM seems like a fantastic system that could be widely adopted, but no details are given on how a lab could build ARM, thus its usefulness is limited.

The reviewer raises a good point, unfortunately the authors are constrained by university policies around patent law. That said efforts are being made to make the ARM widely available to interested researchers. As mentioned above to Reviewer 1’s comments, we end the discussion section with this sentence:

“To promote wide adaptation of this device across as many labs as possible, a company, Tactorum Inc, has been formed.”

(2) The ARM system appears to stop short of hitting the desired forces that von Frey filaments are calibrated toward (Figure 2). This may affect the interpretation of results.

The reviewer gives an important observation. We amended the text to include more clarity on the max forces induced, and comments on causes beyond the delivery mechanism. It should be noted that a newly bought fresh set of von Frey’s was used.

“With the same 1.4 and 2 g von Frey filaments Researcher 1 delivered max average forces of 1.5 g and 2.7 g, and Researcher 2 1.35 g and 2.4 g. The ARM delivered average max forces closest to the targeted forces, with 1.36 g and 1.9 g. (Figure 2C) Some of the error observed could be due to the error rate (+/- 0.05 g) in the force gauge and the von Frey set used.”

(3) The authors mention that ARM generates minimal noise; however, if those sounds are paired with stimulus presentation, they could still prompt a withdrawal response. Including some 'catch' trials in an experiment could test for this.

The reviewer makes a very useful suggestion that we incorporated into our carrageenan experiments. This new data can be found in Supplemental Figure 3F.

“For the carrageenan model, three replicates of the force ramp stimulus were delivered to each paw, and catch trials were performed every 3^rd^ trial to test whether the mice would respond to the noise of the ARM alone. During catch trials, the stimulus was delivered to the open air behind the mouse, and any movement within 5 seconds of stimulus delivery was counted as a response. These trials found a 96% response rate in true trials, with only a 7% rate in catch trials, indicating responses were not being driven by device noise.”

(4) The experimental design in Figure 2 is unclear- did each experimenter have their own cohort of 10 mice, or was a single cohort of mice shared? If shared, there's some concern about repeat testing.

Further clarification was added to avoid confusion on the methods used here.

“Separate cohorts of 10 mice were used for ARM and manual delivery, with a week given between each researcher to avoid sensitization.”

(5) In Figure 5 and S4, the order of the legends does not match the order of the graphs. This can be particularly confusing as the color scheme is not colorblind-friendly. Please consider revising the presentation of these figures.

Corrections made where relevant.

**Reviewer #3 (Public review):**
(1) Limited details are provided for statistical tests and inappropriate claims are cited for individual tests. For example, in Figure 2, differences between researchers at specific forces are reported to be supported by a 2-way ANOVA; these differences should be derived from a post-hoc test that was completed only if the independent variable effects (or interaction effect) were found to be significant in the 2-way ANOVA. In other instances, statistical test details are not provided at all (e.g., Figures 3B, 3C, Figure 4, Figure 6G).

We would like to thank the reviewer for pointing out the lack of clarity in the text on these statistical methods. We have added further details across the manuscript and shown below here in order to address this concern.

“Both manual delivery and the ARM produced significant paw withdrawal percentage curves, a standard traditional measurement of mechanical sensitivity in the field (von Frey 1896, Dixon 1980, Chaplan 1994)(Figure 2E), with a 2-way ANOVA and a posthoc Tukey test detecting significant increases in comparing the 3 lower force VFH’s (0.02g, 0.07g, 0.16g) to the 2 highest force VFH’s (1g, 1.4g). This demonstrates that the ARM delivers results comparable to highly experienced researchers. However, a 2-way ANOVA and a posthoc Tukey test found that Researcher 2 elicited a significantly higher (p=0.0008) paw withdrawal frequency than Researcher 1 (Figure S2A) which corresponded with Researcher 2’s higher VFH application time as measured by the force sensor (Figure 2B).”

“Adjustments were then made to the PAWS software to automate the measurement of withdrawal latency based on pose tracking data of the withdrawal response and the trajectory of the stimulus delivery encoded into the ARM. Testing of C57/BL6J (n=15) at baseline found significant decreases in withdrawal latency for pinprick compared to cotton swab stimuli delivered in identical ways by the ARM (Figure 3B) based on a 2-tailed student t-test.”

“Mice injected with carrageenan (n=15) showed elevated shaking behavior (p=0.0385, 2-way ANOVA and a posthoc Tukey test) in response to pinprick stimuli in comparison to measurements at baseline (Figure 3C).”

“Remote habituated mice showed a significant decrease (p=0.0217, 2-way ANOVA) in time to rest over the 3 days (Figure 4B), but no significant differences for any single day. The number of turns was measured for each group during the first 10 minutes of day 1 to act as a baseline, and then from 20 to 30 minutes for each day. Turn counts were then compared as a percentage of the baseline count for each group. This period was chosen as it the period when experiments start after the day of habituation on experimental days. It was found that remote-habituated mice showed significantly less turning on day 2 compared to mice habituated with a researcher present (p=0.024, 2-way ANOVA posthoc Tukey test), and that only the remote-habituated mice showed significantly decreased turning behavior on day 3 compared to day 1 (p=0.0234, 2-way ANOVA posthoc Tukey test) (Figure 4C).”

“Sex-dependent differences were found in reflexive and affective behavioral components of the mouse withdrawal response when a researcher was present versus not for both reactions to innocuous and noxious stimuli. A 2-way ANOVA and a posthoc Tukey test found that cotton swab stimuli elicited increased male mouse reflexive paw withdrawal features, including max paw height (p=0.0413) and max paw velocity (Y-axis) (p=0.0424) when Researcher 1 was present compared to when no researcher was present (Figure 4E-F). Pinprick stimuli (Figure 4H-I) on the other hand led to increased max paw height (p=0.0436) and max paw velocity (Y-axis) (p=0.0406) in male mice compared to female mice when Researcher 1 was present.

Analysis of the shaking behavior elicited by cotton swab and pinprick stimuli found no significant differences in shaking behavior duration (Figure 4SA-B) but found sex-dependent differences in paw distance traveled after the initial withdrawal, including during shaking and guarding behaviors. For cotton swab (Figure 4G) male mice showed significantly increased paw distance traveled compared to female mice when Researcher 2 was present (p=0.0468, 2-way ANOVA posthoc Tukey test) but not when Researcher 2 was present or no researcher was present. Pinprick stimuli also elicited sex-based increases in paw distance traveled (Figure 4J) in male mice when Researcher 2 was present compared to both male mice when no researcher was present (p=0.0149, 2-way ANOVA posthoc Tukey test) and female mice when Researcher 1 was present (p=0.0038, 2-way ANOVA posthoc Tukey test).”

(2) In the current manuscript, the effects of the experimenter's presence on both habituation time and aspects of the withdrawal reflex are minimal for Researcher 2 and non-existent for Research 1. This is surprising given that Researcher 2 is female; the effect of experimenter presence was previously documented for male experiments as the authors appropriately point out (Sorge et al. PMID: 24776635). In general, this argument could be strengthened (or perhaps negated) if more than N=2 experiments were included in this assessment.

The reviewer makes an important point regarding this data and the need for further experiments. We designed a new set of experiments to examine the effect of male and female researchers overall. It should be noted that this is rather noisy data given it was collected by three sets of male and female researchers over 3 weeks. That said a significant difference was found between mouse sexes when a male researcher was present. This is consistent with previous data, but as we discuss this does not invalidate previous data as researcher gender appears to be only one of the factors at work in researcher presence effects on mouse behavior, leading to individuals having the potential for greater or lesser effects than their overall gender. Our new results can be found in Figure 4K.

“These results indicate that researcher presence at baseline can lead to significant differences in reflexive and affective pain behavior. In this case, male mice showed increased behavioral responses to both touch and pain behavior depending on whether the researcher was present. This led to sex differences in the affective and reflexive component of the withdrawal response when a researcher is present, which disappears when no researcher is present, or a different researcher is present. For this set of researchers, the female researcher elicited the greater behavioral effect. This appeared at first to contradict previous findings (Sorge 2024, Sorge 2014), but it was hypothesized that the effect of an individual researcher could easily vary compared to their larger gender group. To test this, 6 new researchers, half male and half female, were recruited and a new cohort of mice (n=15 male, n=15 female) was tested in each of their presence over the course of 3 weeks, controlling for circadian rhythms (Figure 4K). The newly added force ramp stimulus type was used for these experiments, with three replicates per trial, to efficiently measure mechanical threshold in a manner comparable to previous work. It was found that female mice showed significantly decreased mechanical threshold compared to male mice (p=0.034, Šídák's multiple comparisons test and student’s t-test) when a male researcher was present. This did not occur when a female researcher or no researcher was present. In the latter case of slight trend towards this effect was observed, but it was not significant (p=0.21), and may be the result of a single male researcher being responsible for handling and setting up the mice for all experiments.”

“These findings indicate that sex-dependent differences in evoked pain behavior can appear and disappear based on which researcher/s are in the room. There is a trend towards male researchers overall having a greater effect, but individuals may have a greater or lesser effect on mouse behavior, independent of the gender or sex. This presents a confound that must be considered in the analysis of sex differences in pain and touch behavior which may explain some of the variation in findings from different researchers. Together, these results suggest that remote stimulus delivery may be the best way to eliminate variation caused by experimenter presence while making it easier to compare with data from researchers in your lab and others.”

(3) The in vivo BLA calcium imaging data feel out of place in this manuscript. Is the point of Figure 6 to illustrate how the ARM can be coupled to Inscopix (or other external inputs) software? If yes, the following should be addressed: why do the up-regulated and down-regulated cell activities start increasing/decreasing before the "event" (i.e., stimulus application) in Figure 6F? Why are the paw withdrawal latencies and paw distanced travelled values in Figures 6I and 6J respectively so much faster/shorter than those illustrated in Figure 5 where the same approach was used?

Thanks to the reviewer for bringing up this concern. We have included further text discussing this behavioral data and how it compares to previous work in this study.

“Paw height and paw velocity were found to be consistent with data from figures 4E-I (male researcher and male mice) and 5C (stimulus intensity 2.5 and 4.5) for similar data, with slightly elevated measures of paw distance traveled and decreased paw withdrawal latency for the pinprick stimulus. This was likely caused by sensitization due to multiple stimulus deliveries over the course of the experiment, as due to logistics, 30 stimulus trials were delivered per session due to logistical constraints vs the max of 3 that were performed during previous experiments.”

“This data indicates that the ARM is an effective tool for efficiently correlating in vivo imaging data with evoked behavioral data, including sub-second behavior. One limitation is that the neural response appears to begin slightly before stimulus impact (Figure 6F, 6SB). This was likely caused by a combination of the imprecise nature of ARM v1 paw contact detection and slight delays in the paw contact signal reaching the Inscopix device due to flaws in the software and hardware used, slowing down the signal. Improvements have been made to eliminate this delay as part of the ARM v2, which have been shown to eliminate this delay in vivo fiber photometry data recorded as part of new projects using the device.”

(4) Another advance of this manuscript is the integration of a 500 fps camera (as opposed to a 2000 fps camera) in the PAWS platform. To convince readers that the use of this more accessible camera yields similar data, a comparison of the results for cotton swabs and pinprick should be completed between the 500 fps and 2000 fps cameras. In other words, repeat Supplementary Figure 3 with the 2000 fps camera and compare those results to the data currently illustrated in this figure.

The reviewer makes a good point about the need for direct comparison between 500 fps and 2000 fps data. To address this we added data from same mice, from 2 weeks prior with a comparable set up. These new results can be found in Supplemental Figure 3.

“Changes were made to PAWS to make it compatible with framerates lower than 2000 fps. This was tested using a 0.4 MP, 522 FPS, Sony IMX287 camera recording at 500 fps, and data recorded at 2000 fps by the previously used photron fastcam (Figure 3SC-F). The camera paired with PAWS was found to be sufficient to separate between cotton swab and pinprick withdrawal responses, suggesting it may be a useful tool for labs that cannot invest in a more expensive device. PAWS features measured from 500 fps video data were not significantly different from the 2000 fps data based on a 2 way ANOVA.”

(5) In Figure 2F, the authors demonstrate that a von Frey experiment can be completed much faster with the ARM vs. manually. I don't disagree with that fact - the data clearly show this. I do, however, wonder if the framing of this feature is perhaps too positive; many labs wait > 30 s between von Frey filament applications to prevent receptive field sensitization. The fact that an entire set of ten filaments can be applied in < 50 s (< 3 s between filaments given that each filament is applied for 2 s), while impressive, may never be a feature that is used in a real experiment.

The reviewer makes an important point about how different researchers perform these tests and the relevant timings. We have moderated the framing of these results to address this concern.

“Further, we found that the ARM decreased the time needed to apply a stimulus 10 times to a mouse paw by 50.9% compared to manual delivery (Figure 2F). This effect size may decrease for researchers who leave longer delays between stimulus delivery, but the device should still speed up experiments by reducing aiming time and allowing researchers to quickly switch to a new mouse while waiting for the first.”

(6) Why are different affective aspects of the hindpaw withdrawal shown in different figures? For example, the number of paw shakes is shown in Figure 3C, whereas paw shaking duration is shown in Figure 5D. It would be helpful - and strengthen the argument for either of these measures as being a reproducible, reliable measure of pain - if the same measure was used throughout.

Thanks to the reviewer for pointing out this discrepancy. We have adjusted the figures and text to only use the Number of Paw Shakes for better consistency (Figure 5D and Figure 5-figure supplement 1C).

(7) Is the distance the paw traveled an effective feature of the paw withdrawal (Figure 5E)? Please provide a reference that supports this statement.

A relevant citation and discussion of this metric based on previous studies has been added.

“Mice injected with carrageenan (n=15) showed elevated shaking behavior (p=0.0385) in response to pinprick stimuli in comparison to measurements at baseline (Figure 3C). This aligned with previous findings where PAWS has detected elevations in shaking and/or guarding behavior, examples of affective pain behavior, and post-peak paw distance traveled, which correlates with these behaviors in carrageenan pain models and has been to found to be a good measure of them in past studies (Bohic et al. 2023).”

(8) Dedek et al. (PMID: 37992707) recently developed a similar robot that can also be used to deliver mechanical stimuli. The authors acknowledge this device's ability to deliver optogenetic and thermal stimuli but fail to mention that this device can deliver mechanical stimuli in a similar manner to the device described in this paper, even without experimenter targeting. Additional discussion of the Dedek et al. device is warranted.

We would like to thank the reviewer for identifying this omission. Discussion of this as well as further discussion of Dedek et al.’s automation prototyping work has been added.

“Previous attempts at automating mechanical stimulus delivery, including the electronic von Frey (Martinov 2013) and dynamic plantar asthesiometer (Nirogi 2012), have focused on eliminating variability in stimulus delivery. In contrast to the ARM, both of these devices rely upon a researcher being present to aim or deliver the stimulus, can only deliver vFH-like touch stimuli, and only measure withdrawal latency/force threshold. Additionally, progress has been made in automating stimulus assays by creating devices with the goal of delivering precise optogenetic and thermal stimuli to the mouse’s hind paw (Dedek 2023, Schorscher-Petchu 2021). The Prescott team went farther and incorporated a component into their design to allow for mechanical stimulation but this piece appears to be limited to a single filament type that can only deliver a force ramp. As a result these devices and those previously discussed lack of customization for delivering distinct modalities of mechanosensation that the ARM allows for. Moreover, in its current form the automated aiming of some of these devices may not provide the same resolution or reliability of the ARM in targeting defined targets (Figure 1C), such as regions of the mouse paw that might be sensitized during chronic pain states. Due to the nature of machine learning pose estimation, substantial work, beyond the capacity of a single academic lab, in standardizing the mouse environment and building a robust model based on an extensive and diverse training data set will be necessary for automated aiming to match the reliability or flexibility of manual aiming. That said, we believe this work along with that of that of the other groups mentioned has set the groundwork from which a new standard for evoked somatosensory behavior experiments in rodents will be built.”

(9) Page 2: von Frey's reference year should be 1896, not 1986.

This typo has been fixed, thanks to the reviewer for noting it.

“For more than 50 years, these stimuli have primarily been the von Frey hair (vFH) filaments that are delivered to the mouse paw from an experimenter below the rodent aiming, poking, and subsequently recording a paw lift (von Frey 1896, Dixon 1980, Chaplan 1994).”

(10) Page 2: Zumbusch et al. 2024 also demonstrated that experimenter identification can impact mechanical thresholds, not just thermal thresholds.

Text has been updated in order to note this important point.

“A meta-analysis of thermal and mechanical sensitivity testing (Chesler 2002, Zumbusch 2024) found that the experimenter has a greater effect on results than the mouse genotype, making data from different individual experimenters difficult to merge.”

(11) Page 2: One does not "deliver pain in the periphery". Noxious stimuli or injury can be delivered to the periphery, but by definition, pain is a sensation that requires a central nervous system.

Text has been updated for improved accuracy.

“Combining approaches to deliver painful stimuli with techniques mapping behavior and brain activity could provide important insights into brain-body connectivity that drives the sensory encoding of pain.”